# The retromer complex safeguards against neural progenitor-derived tumorigenesis by regulating Notch receptor trafficking

Bo Li[1†], Chouin Wong[1†], Shihong Max Gao[1], Rulan Zhang[1], Rongbo Sun[2,3], Yulong Li[2,3,4], Yan Song[1,4]*

[1]Ministry of Education Key Laboratory of Cell Proliferation and Differentiation, School of Life Sciences, Peking University, Beijing, China; [2]State Key Laboratory of Membrane Biology, School of Life Sciences, Peking University, Beijing, China; [3]PKU-IDG/McGovern Institute for Brain Research, Beijing, China; [4]Peking-Tsinghua Center for Life Sciences, Peking University, Beijing, China

**Abstract** The correct establishment and maintenance of unidirectional Notch signaling are critical for the homeostasis of various stem cell lineages. However, the molecular mechanisms that prevent cell-autonomous ectopic Notch signaling activation and deleterious cell fate decisions remain unclear. Here we show that the retromer complex directly and specifically regulates Notch receptor retrograde trafficking in *Drosophila* neuroblast lineages to ensure the unidirectional Notch signaling from neural progenitors to neuroblasts. Notch polyubiquitination mediated by E3 ubiquitin ligase Itch/Su(dx) is inherently inefficient within neural progenitors, relying on retromer-mediated trafficking to avoid aberrant endosomal accumulation of Notch and cell-autonomous signaling activation. Upon retromer dysfunction, hypo-ubiquitinated Notch accumulates in Rab7$^+$ enlarged endosomes, where it is ectopically processed and activated in a ligand-dependent manner, causing progenitor-originated tumorigenesis. Our results therefore unveil a safeguard mechanism whereby retromer retrieves potentially harmful Notch receptors in a timely manner to prevent aberrant Notch activation-induced neural progenitor dedifferentiation and brain tumor formation.
DOI: https://doi.org/10.7554/eLife.38181.001

*For correspondence:
yan.song@pku.edu.cn

†These authors contributed equally to this work

Competing interests: The authors declare that no competing interests exist.

## Introduction

The correct establishment and maintenance of unidirectional Notch signaling are critical for the homeostasis of various stem cell lineages (*Bertet et al., 2014*; *Blanpain et al., 2006*; *Bowman et al., 2008*; *Conboy and Rando, 2002*; *Fre et al., 2005*; *Guo and Ohlstein, 2015*; *Lin et al., 2012*; *Liu et al., 2017*; *Ohlstein and Spradling, 2007*; *Ren et al., 2018*; *Song and Lu, 2011*; *Williams et al., 2011*). The canonical Notch signaling, which requires two adjacent cells to present transmembrane ligands and transmembrane receptors respectively (*Bray, 2006*; *Kopan and Ilagan, 2009*) and involves intercellular or intracellular amplification step(s) to establish its unidirectionality (*Artavanis-Tsakonas et al., 1999*; *Liu et al., 2017*; *Losick and Desplan, 2008*), is an ideal signaling pathway for binary cell fate specification. Accordingly, Notch signaling has been implicated in cell fate decision-making events in diverse stem cell lineages (*Bertet et al., 2014*; *Chen et al., 2016*; *Demitrack et al., 2015*; *Dong et al., 2012*; *Hilton et al., 2008*; *Homem and Knoblich, 2012*; *Liu et al., 2010*; *Ohlstein and Spradling, 2007*; *Pinto-Teixeira et al., 2018*; *Watt et al., 2008*).

An important strategy utilized by dividing stem cells or progenitors to ensure binary cell fate decisions is asymmetric segregation of the endocytic protein Numb, an evolutionarily conserved Notch signaling antagonist, to one of the daughter cells (*Bultje et al., 2009*; *Conboy and Rando, 2002*;

**eLife digest** Most cells in the animal body are tailored to perform particular tasks, but stem cells have not yet made their choice. Instead, they have unlimited capacity to divide and, with the right signals, they can start to specialize to become a given type of cells. In the brain, this process starts with a stem cell dividing. One of the daughters will remain a stem cell, while the other, the neural progenitor, will differentiate to form a mature cell such as a neuron. Keeping this tight balance is crucial for the health of the organ: if the progenitor reverts back to being a stem cell, there will be a surplus of undifferentiated cells that can lead to a tumor.

A one-way signal driven by the protein Notch partly controls the distinct fates of the two daughter cells. While the neural progenitor carries Notch at its surface, its neural stem cell sister has a Notch receptor on its membrane instead. This ensures that the Notch signaling goes in one direction, from the cell with Notch to the one sporting the receptor.

When a stem cell divides, one daughter gets more of a protein called Numb than the other. Numb pulls Notch receptors away from the external membrane and into internal capsules called endosomes. This guarantees that only one of the siblings will be carrying the receptors at its surface. Yet, sometimes the Notch receptors can get activated in the endosomes, which may make neural progenitors revert to being stem cells. It is still unclear what tools the cells have to stop this abnormal activation.

Here, Li et al. screened brain cells from fruit fly larvae to find out the genes that might play a role in suppressing the inappropriate Notch signaling. This highlighted a protein complex known as the retromer, which normally helps to transport proteins in the cell. Experiments showed that, in progenitors, the retromer physically interacts with Notch receptors and retrieves them from the endosomes back to the cell surface. If the retromer is inactive, the Notch receptors accumulate in the endosomes, where they can be switched on. It seems that, in fruit flies, the retromer acts as a bomb squad that recognizes and retrieves potentially harmful Notch receptors, thereby preventing brain tumor formation.

Several retromer components are less present in patients with various cancers, including glioblastoma, an aggressive form of brain cancer. The results by Li et al. may therefore shed light on the link between the protein complex and the emergence of the disease in humans.

DOI: https://doi.org/10.7554/eLife.38181.002

*Gunage et al., 2014*; *Lu et al., 1998*; *Luo et al., 2005*; *Rhyu et al., 1994*; *Sallé et al., 2017*; *Shen et al., 2002*; *Wang et al., 2006*; *Wu et al., 2017*; *Zhong et al., 1996*). Numb acts as an adaptor to bridge the Notch receptor and its cofactor(s) with the endocytic machinery and reduces the surface pool of Notch by promoting its endocytosis (*Hutterer and Knoblich, 2005*; *Song and Lu, 2012*). Endocytosed Notch receptors are often poly-ubiquitinated by E3 ubiquitin ligases, such as Itch/Su(dx) (Suppressor of deltex) and Nedd4 (*Cornell et al., 1999*; *Le Bras et al., 2011*; *Qiu et al., 2000*; *Sakata et al., 2004*; *Wilkin et al., 2004*), and sorted through the ESCRT (Endosomal Sorting Complex Required for Transport) pathway for lysosomal degradation (*Horner et al., 2018*; *Thompson et al., 2005*; *Vaccari et al., 2009*). As a consequence, the daughter cell inheriting relatively more Numb protein becomes the Notch signaling sending cell, unambiguously establishing signaling directionality. Not surprisingly, dysregulation in the asymmetric segregation of Numb has been implicated in a wide range of developmental defects and diseases (*Bowman et al., 2008*; *Bu et al., 2016*; *Caussinus and Gonzalez, 2005*; *George et al., 2013*; *Li et al., 2003*; *Pece et al., 2004*).

However, the plasma membrane is not the only location where the Notch receptor can be processed and activated. The proteolytic activity of γ-secretase has been detected in endosomal membranes (*Gupta-Rossi et al., 2004*; *Lah and Levey, 2000*; *Pasternak et al., 2003*; *Urra et al., 2007*). Furthermore, it has been postulated that the relatively low pH at the endosomal compartments renders a conformational change in the Notch receptor, allowing for more efficient proteolysis. Indeed, inactivation of the ESCRT complex leads to retention of the Notch receptor in the limiting membrane of multivesicular bodies (MVBs) where Notch is ectopically activated via ligand-independent, γ-secretase-dependent proteolysis (*Hori et al., 2011*; *Thompson et al., 2005*; *Vaccari and Bilder,*

*2005*; *Vaccari et al., 2009*; *Wilkin et al., 2008*; *Zhou et al., 2016*). Other than ESCRT pathway-mediated lysosomal degradation, how protein trafficking machinery prevents deleterious cell-autonomous Notch signaling activation in stem cell lineages remains to be elucidated.

Type II neural stem cells, so called neuroblasts, in the *Drosophila* larval central brain region provide an attractive model system for studying how endosomal trafficking establishes unidirectional Notch signaling and ensures stem cell versus progenitor binary cell fate decisions (*Figure 1A*) (*Liu et al., 2017*; *Song and Lu, 2012*). Firstly, type II neural stem cell lineages resemble their mammalian counterparts in terms of regulatory molecules and principles, yet with much simpler anatomical structure and lineage composition (*Brand and Livesey, 2011*; *Homem and Knoblich, 2012*; *Sousa-Nunes et al., 2010*). Secondly, unidirectional Notch signaling is critical for establishing type II neuroblast versus immature intermediate neural progenitor (INP) binary cell fates (*Bowman et al., 2008*; *Song and Lu, 2011*; *Song and Lu, 2012*; *Wang et al., 2006*; *Weng et al., 2010*). Whereas downregulation of Notch signaling in neuroblasts leads to their premature differentiation into INPs and loss of stemness, overactivation of Notch signaling in neural progenitors cause their fate reversion back into neuroblast-like state and tumorigenesis (*Bowman et al., 2008*; *Song and Lu, 2011*; *Song and Lu, 2012*; *Wang et al., 2006*; *Weng et al., 2010*). Thus, the total number of neuroblasts in each brain lobe represents a quantitative and precise readout of Notch signaling strength. Thirdly, Numb is asymmetrically inherited by immature INPs, where it dampens Notch signaling partly by reducing the cell surface pool of mature Notch receptors (*Figure 1B*) (*Bowman et al., 2008*; *Lee et al., 2006b*; *Song and Lu, 2012*; *Wang et al., 2006*).

In a large-scale unbiased RNAi-based genetic screen for regulators of neuroblast versus progenitor cell fate decision, we identified Vps26, a subunit of the retromer complex (*Burd and Cullen, 2014*; *Wang and Bellen, 2015*). Specific downregulation of Vps26 in *Drosophila* central brain neuroblast lineages led to a supernumerous neuroblast phenotype. The retromer complex is an evolutionarily highly conserved endosomal sorting complex, which plays a crucial role in the retrograde trafficking of a specific subset of endocytosed proteins from endosomes back to the trans-Golgi network or the plasma membrane (*Burd and Cullen, 2014*; *Wang and Bellen, 2015*). The core of the retromer complex is a vacuolar protein sorting (Vps) trimer composed of Vps35, Vps26 and Vps29 subunits (*Figure 1C*). Previous studies have implicated retromer in controlling a wide range of physiological processes, such as regulating fly wing development, maintaining the function of photoreceptors, establishing cell polarity in epithelial cells, controlling LTP (long-term potential) in mature hippocampus, modulating fly oogenesis and propagating mitochondrial stress signals (*Belenkaya et al., 2008*; *Chen et al., 2010*; *Choy et al., 2014*; *Coudreuse et al., 2006*; *Franch-Marro et al., 2008*; *Gomez-Lamarca et al., 2015*; *Harterink et al., 2011*; *Hesketh et al., 2014*; *Pan et al., 2008*; *Pocha et al., 2011*; *Port et al., 2008*; *Starble and Pokrywka, 2018*; *Temkin et al., 2011*; *Temkin et al., 2017*; *Wang and Bellen, 2015*; *Yang et al., 2008*; *Zhang et al., 2018*). Dysfunction of retromer-mediated endosomal sorting has been linked to various pathologies, including neurodegenerative diseases such as Alzheimer's disease and Parkinson's disease (*McMillan et al., 2017*; *Small and Petsko, 2015*; *Wang and Bellen, 2015*).

Here our results unveil a safeguard mechanism through which the retromer complex ensures sufficient dampening of Notch signaling in neural progenitors. Upon attenuation of the retromer function, hypo-ubiquitinated Notch that fails to enter the ESCRT-lysosomal pathway accumulates in enlarged Rab7[+] endosomes and is ectopically processed and activated. Such cell-autonomous intracellular hyperactivation of Notch signaling causes fate reversion of neural progenitors and the formation of transplantable tumors. These results led us to propose a model whereby retromer serves as 'bomb squad' to retrieve and disarm the potentially harmful pool of Notch receptors in a timely manner.

## Results

### The retromer complex prevents neural progenitor dedifferentiation and tumorigenesis

To investigate the function of retromer in neuroblast lineages, we first downregulated Vps26 in all central brain neuroblast lineages using short hairpin microRNAs (shmiRNAs), driven by *insc*-Gal4, and observed a supernumerary neuroblast phenotype (*Figure 1D,E*). Such brain tumor phenotype

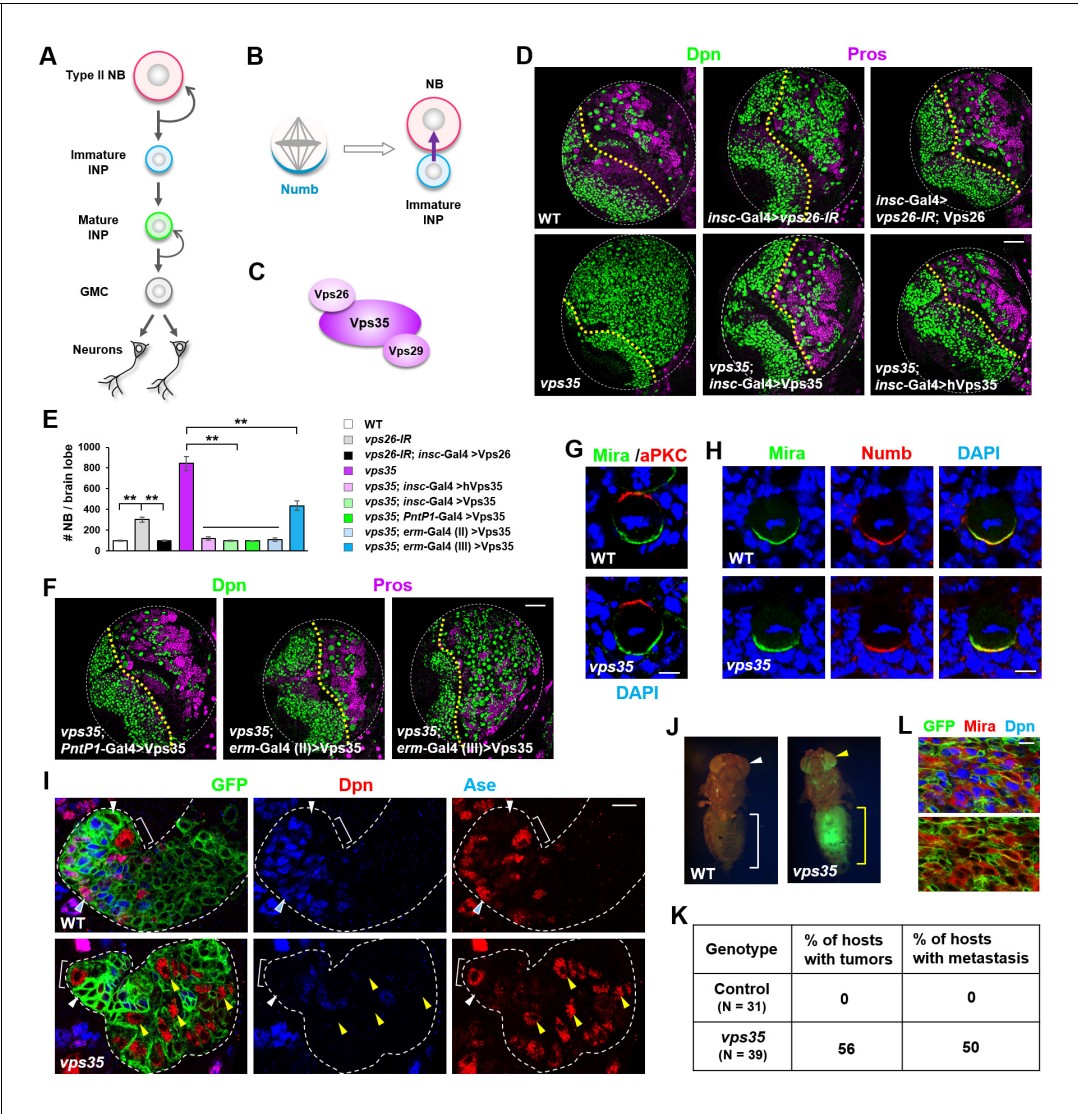

**Figure 1.** Dedifferentiation of *vps35* mutant neural progenitors causes the formation of transplantable tumors. (A) Diagram depicting the lineage hierarchy of *Drosophila* type II neuroblasts in the central brain area. (B) Schematic showing how asymmetric distribution and segregation of the endocytic protein Numb (cyan) initiates unidirectional Notch signaling (purple arrow) from a neural progenitor (light blue) to its sibling type II neuroblast (pink). (C) Schematic of the cargo-recognition retromer complex. (D–F) Larval brain lobes of indicated genotypes were stained for neuroblast marker Deadpan (Dpn) and ganglion mother cell (GMC)/neuronal marker Prospero (nuclear Pros) (D,F). In this and subsequent micrographs, yellow dotted line marks the boundary between the optic lobe (left) and the central brain (right) areas. Quantification of total neuroblast number per brain lobe is shown in (E). **p<0.001 (n = 12–16). (G) Asymmetric cortical distribution of apical marker atypical PKC (aPKC) and basal marker Miranda (Mira) in wild type (WT) or *vps35* mutant metaphase neuroblasts. (H) Colocalization of Mira and cell fate determinant Numb at the basal cortex of WT or *vps35* mutant metaphase neuroblasts. (I) MARCM clonal analysis of type II neuroblast lineages in WT control or *vps35* mutant backgrounds. In this and subsequent micrographs, type II neuroblast MARCM clones are marked by CD8-GFP and outlined by white dashed lines, whereas neuroblasts, immature intermediate neural progenitors (INPs), mature INPs and neuroblast-like dedifferentiating progenitors are marked with brackets, white arrowheads, cyan arrowheads and yellow arrowheads respectively. (J) Transplantation of GFP$^+$ tissue from WT control larval brains into the abdomens of adult host flies caused neither tumorous growth (while bracket) nor metastasis (white arrowhead). In sharp contrast, transplantation of GFP$^+$ tumor tissue from *vps35* mutant larval brains caused massive tumor formation (yellow bracket) and metastasis to distal organs such as the eyes (yellow arrowhead). (K) Table showing the frequency of tumor formation or metastasis 14 days after transplantation of GFP$^+$ tissue from larval brains of indicated genotypes. (L) GFP$^+$ tumor tissues from the transplanted hosts were isolated and stained for neuroblast markers Mira and Dpn. Note that most of the extracted GFP$^+$ tumor cells were Mira$^+$ and Dpn$^+$ neuroblast-like cells. Scale bars, 50 µm (D,F); 5 µm (G,H) and 10 µm (I,L).

DOI: https://doi.org/10.7554/eLife.38181.003

The following source data and figure supplements are available for figure 1:

**Source data 1.** Input data for bar graph *Figure 1E*.

*Figure 1 continued on next page*

*Figure 1 continued*

DOI: https://doi.org/10.7554/eLife.38181.008

**Figure supplement 1.** A summary of the Gal4 drivers and cell type markers used in this study.

DOI: https://doi.org/10.7554/eLife.38181.004

**Figure supplement 2.** Ectopic neuroblasts formed upon retromer dysfunction are originated from immature INPs.

DOI: https://doi.org/10.7554/eLife.38181.005

**Figure supplement 3.** MARCM clonal analysis of wild type and *vps35* mutant neuroblasts.

DOI: https://doi.org/10.7554/eLife.38181.006

**Figure supplement 4.** Lineage-tracing analysis of wild type and *vps35* mutant immature neural progenitors.

DOI: https://doi.org/10.7554/eLife.38181.007

induced by *vps26-RNAi* was fully rescued by the coexpression of a shmiRNA-resistant form of the Vps26 transgene, excluding the possibility of an off-target effect of the shmiRNA (*Figure 1D,E*). Furthermore, homozygous *vps35* mutant larval brains exhibited an even more severe supernumerary neuroblast phenotype than *vps26-IR*, and such phenotype was fully rescued upon specific expression of a Vps35 transgene in all central brain neuroblast lineages (*Figure 1D,E*). Importantly, human Vps35 also fully rescued the brain tumor phenotype of *vps35* mutants back to wild type (*Figure 1D, E*). Taken together, our results clearly indicated that retromer plays an evolutionarily-conserved role in preventing ectopic neuroblast formation in the central brain area.

To investigate the cellular origin of the ectopic neuroblasts formed upon retromer inactivation, we expressed the Vps35 transgene in distinct subset of cells within central brain neuroblast lineages and assessed its ability to rescue the *vps35* mutant phenotype. Expression of the Vps35 transgene in type II neuroblast lineages, by *PntP1*-Gal4 (*Zhu et al., 2011*), fully suppressed the brain tumor phenotype caused by *vps35* mutation (*Figure 1E,F* and *Figure 1—figure supplement 1*). By contrast, restoring Vps35 function in type I neuroblast lineages, by *ase*-Gal4 (*Zhu et al., 2006*), failed to do so (data not shown). These results indicated that the ectopic neuroblasts in retromer mutants are derived from type II neuroblast lineages. Indeed, expression of the Vps35 transgene in Deadpan (Dpn)$^-$ Asense (Ase)$^-$ INPs by *erm*-Gal4 (II) but not in Dpn$^-$ Ase$^+$ INPs by *erm*-Gal4 (III) (*Pfeiffer et al., 2008*) completely rescued the supernumerary neuroblast phenotype caused by Vps35 inactivation (*Figure 1E,F* and *Figure 1—figure supplement 1*). Therefore, the reverting Dpn$^-$Ase$^-$ neural progenitors are the cellular origin of brain tumor in *vps35* mutants. Supporting this notion, cell polarity remained unaltered in *vps35* mutant neuroblasts (*Figure 1G*), indicating that these ectopic neuroblasts are not resulted from neuroblast symmetric division. Importantly, Numb is normally localized to the basal cortex of *vps35* mutant dividing neuroblasts (*Figure 1H*), arguing against the possibility that defective asymmetric segregation of Numb causes INP dedifferentiation in *vps35* mutant brains. Consistently, whereas Vps26 downregulation in type II neuroblast lineages or immature INP lineages, driven by *PntP1*-Gal4 or *erm*-Gal4(II) respectively, resulted in supernumerary neuroblast phenotype, its knockdown in mature INP lineages or type I neuroblast lineages, driven by *erm*-Gal4 (III) or *ase*-Gal4 respectively, failed to induce ectopic neuroblasts (*Figure 1—figure supplement 2*). Furthermore, distinct from wild type control type II neuroblast MARCM clones (*Lee and Luo, 1999*) that contained one and only one Dpn$^+$ Ase$^-$ neuroblast (white bracket in *Figure 1I* and *Figure 1—figure supplement 1B*), *vps35* mutant clones contained multiple ectopic Dpn$^+$ Ase$^-$ Pros$^-$ neuroblast-like cells (yellow arrowheads in *Figure 1* and *Figure 1—figure supplement 3*) several cell diameters away from the primary neuroblast (white bracket in *Figure 1I*). These ectopic Dpn$^+$ Ase$^-$ Pros$^-$ neuroblast-like cells were of intermediate cell sizes between neural progenitors and primary neuroblasts (yellow arrowheads in *Figure 1* and *Figure 1—figure supplement 3*), indicating that they were undergoing dedifferentiation (*Song and Lu, 2011*). In addition, FLP-FRT-based lineage tracing by inducing GFP$^+$ clones exclusively in immature INPs, driven by *erm*-Gal4 (II), resulted in labeling of INPs (white arrowhead in *Figure 1—figure supplement 4*), GMCs, and neurons (cyan arrowhead in *Figure 1—figure supplement 4*) in wild-type brains. In contrast, in *vps35* mutant brains, GFP-labeled ectopic type II neuroblasts of various cellular sizes were found after similar lineage tracing (yellow arrowheads in *Figure 1—figure supplement 4*), indicating that immature INPs could indeed dedifferentiate back into neuroblast-like cells upon retromer dysfunction. Taken together, our results clearly indicate that the brain tumor phenotype in *vps35* mutants is caused by cell fate reversion of Dpn$^-$ Ase$^-$ neural progenitors.

We next employed transplantation assay to test whether the ectopic neuroblasts in *vps35* mutant brains are capable of initiating tumor. Transplantation of *vps35* mutant but not wild-type control brain tissues into the abdomens of host flies caused the formation of massive tumors (yellow bracket in *Figure 1J*) that often metastasize to distal organs (yellow arrowhead in *Figure 1J*; statistic results in *Figure 1K*). Importantly, the *vps35* mutant GFP[+] tumor cells extracted from the abdomen of transplanted hosts were Dpn[+] Miranda (Mira)[+] neuroblast-like cells (*Figure 1L*). Thus *vps35* mutant cells in the larval brains are indeed tumor-initiating cells. Together, we conclude that retromer acts as a tumor suppressor in the *Drosophila* brain by preventing neural progenitor dedifferentiation.

## *vps35* mutant dedifferentiating neural progenitors contained enlarged Rab7-positive endosomal vesicles

Since the well-characterized function of retromer is retrograde transport of transmembrane proteins, we next assessed whether the distribution of any subcellular marker(s) is altered upon inactivation of retromer function. Compared to wild-type control INPs, *vps35* mutant INPs or ectopic neuroblasts displayed dramatically enlarged late endosomes/MVBs (*Figure 2A–C*; up to more than 10-fold increase in endosomal vesicle sizes). The expression levels of Rab7 remained unchanged in *vps35* mutants (*Figure 2D*), ruling out the likelihood that Vps35 regulates Rab7 gene expression or protein stability. Furthermore, Rab7 primarily colocalized with early endosome marker Rab5 in *vps35* but not

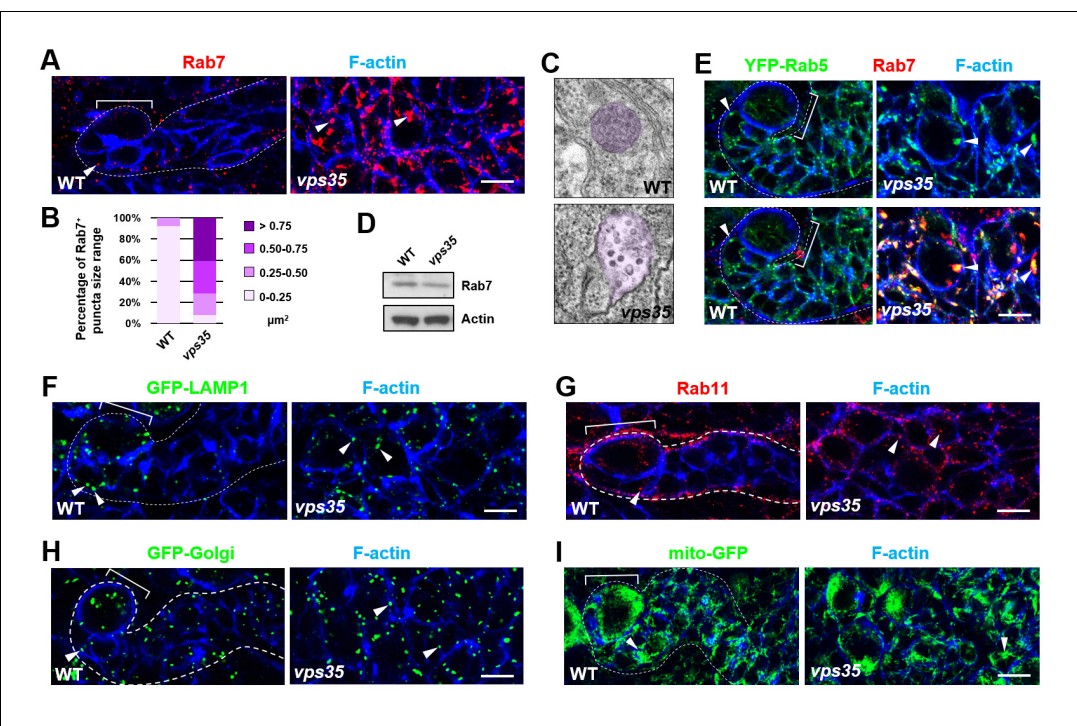

**Figure 2.** Rab7[+] endosomes are drastically enlarged in *vps35* mutant neuroblast lineages. (**A,B**) Compared to WT control immature INPs, Rab7[+] endosomes were dramatically enlarged in *vps35* mutant dedifferentiating neural progenitors (arrowheads in **A**). Quantification of the size range of Rab7[+] puncta in immature INPs of indicated genotypes was shown in (**B**). (**C**) Transmission electron micrograph of wild type or *vps35* mutant larval brain neuroblasts. The mean size of MVBs, identified by the presence of intraluminal vesicles, was greatly enlarged in *vps35* mutant neuroblasts. Note that neuroblasts were identified by their large cellular and nuclear sizes and MVBs are highlighted in purple. (**D**) Western blot analysis of larval brain extracts of indicated genotypes using anti-Rab7 antibody. Anti-β-actin blot served as a loading control. (**E**) The enlarged Rab7[+] endosomes in *vps35* mutant neuroblast-like cells were also positive for YFP-Rab5 (arrowheads). (**F–I**) Compared to WT control immature INPs, the sizes of GFP-LAMP1[+] lysosomes (**F**), Rab11[+] recycling endosomes (**G**), Golgi (**H**) or mitochondria marked by mito-GFP (**I**) remained unaltered in *vps35* mutant dedifferentiating neural progenitors (arrowheads). Scale bars, 10 μm (**A,E–I**).
DOI: https://doi.org/10.7554/eLife.38181.009

wild type cells (*Figure 2E*), demonstrating that the enlarged MVBs in *vps35* mutant cells are of early and late endosome hybrid identities. In contrast, other subcellular markers including lysosome (GFP-LAMP1), recycling endosome (Rab11), Golgi (GFP-Golgi) and mitochondria (mito-GFP) remained unchanged in *vps35*-defective cells (*Figure 2F–2I*). Therefore, our results strongly suggest that retromer normally functions in neural progenitors to transport cargo proteins away from early and late endosomes. Upon retromer dysfunction, its cargo proteins highly accumulate in MVBs, resulting in enlarged, aberrant endosomal vesicles of hybrid identities.

## Retromer regulates retrograde trafficking of Notch receptors

We next sought to identify the critical cargo protein(s) of retromer in preventing INP dedifferentiation. Since Notch pathway is both necessary and sufficient to promote self-renewal in type II neuroblast lineages, we first examined the subcellular distribution of transmembrane protein components of Notch signaling pathway. We noted that the Notch receptor and its cofactor Sanpodo (*Couturier et al., 2012*; *Hutterer and Knoblich, 2005*; *O'Connor-Giles and Skeath, 2003*; *Song and Lu, 2012*) highly accumulated in enlarged puncta in *vps35* mutant cells, mostly colocalizing with Rab7$^+$ enlarged endosomes (*Figure 3A,B* and *Figure 3—figure supplement 1*). In contrast, the distribution of other signaling molecules such as Patched (Ptc) and Wnt/Wingless (Wg) remained unaltered upon Vps35 depletion (*Figure 3—figure supplement 2A–C*), indicating that retromer specifically mediates Notch receptor trafficking in neuroblast lineages. Strongly supporting this notion, Notch signaling reporter E(spl)mγ-GFP (*Almeida and Bray, 2005*; *Song and Lu, 2011*), which faithfully reflects Notch signaling activity in neuroblast lineages, was undetectable in wild type Dpn$^-$ Ase$^-$ immature INPs (white arrowhead in *Figure 3C*) but ectopically turned on in Dpn$^+$ Ase$^-$ dedifferentiating neural progenitors (yellow arrowhead in *Figure 3C*) upon Vps26 downregulation. In addition, Notch puncta colocalizing with Rab7$^+$ endosomes remained unaltered in *vps35* mutant wing imaginal disc epithelia (arrowheads in *Figure 3—figure supplement 2D*), suggesting a tissue-specific regulation of Notch trafficking by retromer. Collectively, retromer normally suppresses Notch activity through mediating retrograde trafficking of Notch receptors in neural progenitors.

We next assessed whether Notch is a crucial cargo of retromer in neuroblast lineages. Neuroblast lineage-specific knockdown of Notch completely suppressed the neuroblast overproliferation phenotype in *vps35* mutants (*Figure 3D,E*), indicating that the dedifferentiation process of *vps35* mutant INPs was Notch signaling-dependent. Type II neuroblast lineage-specific or immature INP-specific depletion of the ligand Delta, as well as neuroblast lineage-specific expression of a dominant negative form of Delta (*Dl-DN*) that lacks its intracellular domain (*Baonza et al., 2000*; *Flores et al., 2000*; *Huppert et al., 1997*), completely or potently suppressed brain tumor phenotypes caused by *vps35* mutations (*Figure 3D,E* and *Figure 3—figure supplement 2E,F*). Furthermore, type II neuroblast lineage-specific or immature INP-specific expression of a dominant negative form of the metalloprotease Kuzbanian (*Kuz-DN*), which lacks its protease activity and thereby specifically blocks ligand-induce S2 cleavage of Notch (*Lieber et al., 2002*; *Mumm et al., 2000*; *Pan and Rubin, 1997*), also phenocopied the effect of *Notch-RNAi* in inhibiting brain tumor formation (*Figure 3D, E*). These observations indicated that overactivation of Notch signaling in *vps35* mutant neural progenitors is largely, if not completely, ligand-dependent. Not surprisingly, a functional γ–secretase is also essential for ectopic activation of Notch signaling in *vps35* mutants (*Figure 3D,E*). In sharp contrast, inactivation of various other signaling pathways, such as Wnt/Wg, Hedgehog or EGFR, or overactivation of Hedgehog signaling showed no effects on the supernumerary neuroblast phenotype in *vps35* mutants (*Figure 3D,E* and *Figure 3—figure supplement 2E,F*), further demonstrating the high specificity of retromer on Notch signaling pathway in neuroblast lineages. Importantly, Notch colocalized with fly or human Vps35 transgene (*Figure 3F,G*) and endogenous Vps26 (*Figure 3H* and *Figure 3—figure supplement 3*). More remarkably, Notch depletion by RNAi led to a dramatic reduction in Rab7$^+$ endosomal vesicle sizes almost back to normal (*Figure 3I*), suggesting that Notch receptors constitute the major endosomal contents of these aberrant *vps35* mutant vesicles. Taken together, our results strongly suggested that the Notch receptor is a functionally important cargo of retromer in type II neuroblast lineages.

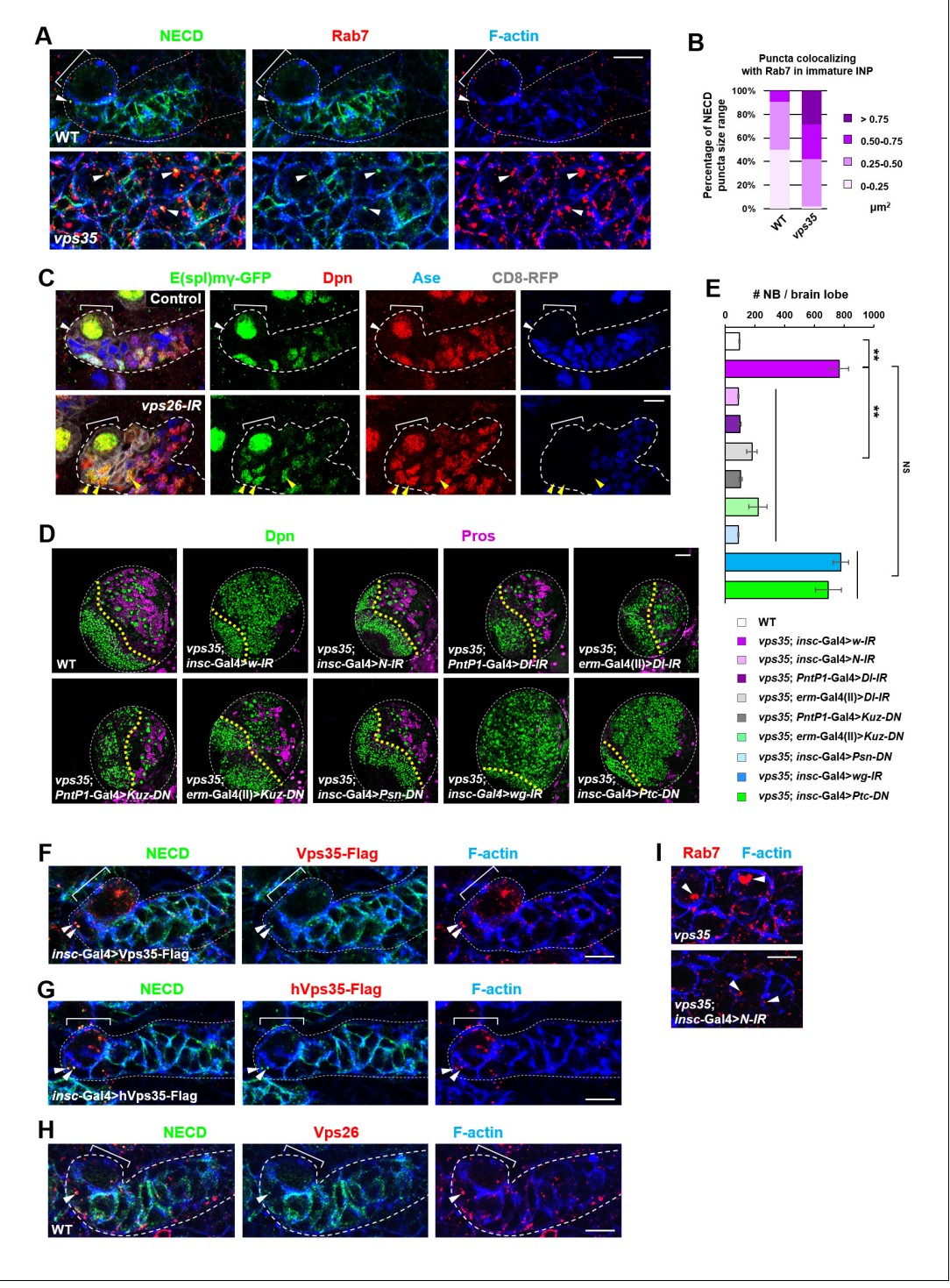

**Figure 3.** Retromer regulates Notch signaling by mediating Notch receptor endosomal trafficking. (**A,B**) Compared to WT control immature INPs, Notch puncta colocalizing with Rab7+ endosomes were enlarged in *vps35* mutant dedifferentiating neural progenitors (arrowheads in **A**). Quantification of the size range of Notch puncta colocalizing with Rab7+ endosomes is shown in (**B**). (**C**) The expression pattern of Notch signaling reporter E(spl)mγ-GFP in wild type control or *vps26-RNAi* type II neuroblast lineages. Note that immature INPs (Dpn⁻ Ase⁻) in control type II neuroblast lineages and dedifferentiating neural progenitors (Dpn⁺ Ase⁻) in *vps26-RNAi* lineages are marked with white arrowheads and yellow arrowheads respectively. (**D,E**) Larval brain lobes of indicated genotypes were stained for Dpn and Pros. Quantification of total neuroblast number per brain lobe is shown in (**E**). **p<0.001 (n = 10–18). NS, not significant. (**F**) Type II neuroblast lineages expressing Vps35-FLAG were stained for

*Figure 3 continued*

Notch extracellular domain (NECD) and FLAG. Note that NECD puncta (arrowheads) colocalized with Vps35-FLAG in immature INPs. (G,H) NECD puncta colocalized with FLAG-tagged human Vps35 (hVps35-FLAG; G) and endogenous Vps26 (H) in immature INPs (arrowheads). (I) Type II neuroblast lineages of indicated genotypes were stained for Rab7 and F-actin. Rab7 puncta are marked with arrowheads. Scale bars, 10 µm (A,C,F–I) and 50 µm (D).

DOI: https://doi.org/10.7554/eLife.38181.010

The following source data and figure supplements are available for figure 3:

**Source data 1.** Input data for bar graph *Figure 3E*.
DOI: https://doi.org/10.7554/eLife.38181.014
**Figure supplement 1.** Retromer regulates endosomal trafficking of Notch cofactor Sanpodo (Spdo).
DOI: https://doi.org/10.7554/eLife.38181.011
**Figure supplement 2.** Retromer regulates Notch signaling in fly neuroblast lineages with high specificity.
DOI: https://doi.org/10.7554/eLife.38181.012
**Figure supplement 3.** Antibody raised against Vps26 is highly specific.
DOI: https://doi.org/10.7554/eLife.38181.013

## Notch is a bona fide cargo protein of retromer

To validate that the Notch receptor is a cargo protein of the retromer complex, we assessed their physical interaction by performing coimmunoprecipitation (coIP) assays. Vps35 or Vps26 was specifically coimmunoprecipitated with Notch intracellular domain (NICD) from HEK293T cell extracts (*Figure 4A*). Further domain mapping experiments revealed that the ankyrin repeat region but not the C-terminal region of NICD exhibited a strong binding activity to Vps26 (*Figure 4B,C*). Reciprocal coIP assay showed that Vps26 utilized its middle domain to interact with NICD (*Figure 4D,E*). Furthermore, Notch-V5 expressed in central brain neuroblast lineages was specifically coimmunoprecipitated with Vps35-FLAG from fly larval brain extracts (*Figure 4F*), confirming the in vivo protein-protein interaction. Importantly, coIP experiments further revealed interaction between mouse NICD and mouse Vps26 proteins (*Figure 4G,H*), indicating that the physical association between the retromer cargo-recognition complex and Notch is evolutionarily conserved. Taken together, our results validate that the Notch receptor is a bona fide cargo protein of the retromer complex.

## Retromer prevents intracellular hyperactivation of Notch signaling

Our results presented so far support an intriguing possibility that the retromer complex physically interacts with Notch and transports it away from early and late endosomes in a timely and efficient manner. When retromer is defective, Notch receptors are trapped at early/late aberrant endosomal vesicles and is ectopically processed and activated, causing neural progenitor-derived brain tumor.

If this hypothesis is correct, one would expect that blocking the flux of Notch receptors towards its activating compartment or accelerating Notch trafficking away from it might prevent the accumulation and subsequent ectopic activation of Notch in *vps35* mutants (*Figure 5A*). Indeed, overexpression of a dominant negative form of Rab5 GTPase (*Rab5-DN*), which blocks the fusion of endocytic vesicles with early endosomes, or a constitutively active form of Rab9 GTPase (Rab9-CA), which promotes protein retrograde trafficking from late endosomes to trans-Golgi network (TGN) or the plasma membrane (*Figure 5—figure supplement 1*), completely suppressed brain tumor formation in *vps35* mutant brains (*Figure 5B,C*). Importantly, both the enlargement of Rab5/Rab7-positive endosomal vesicles and the high accumulation of Notch in these aberrant endosomal compartments in *vps35* mutant cells were effectively relieved upon *Rab5-DN* or Rab9-CA coexpression (arrowheads in *Figure 5D,E*). On the other hand, overexpression of a constitutively active form of Rab7 (Rab7-CA) or the ESCRT-0 complex component Hrs (Hepatocyte growth factor-regulated tyrosine kinase substrate), which accelerates the protein trafficking towards lysosome (*Figure 5—figure supplement 1*) (*Lloyd et al., 2002*), potently inhibited the neuroblast overproliferation phenotype in *vps35* mutant brains (*Figure 5A,B,C*). Indeed, coexpression of either Rab7-CA or Hrs led to high accumulation of Notch in lysosomes of *vps35* mutant cells (*Figure 5F*). In contrast, overexpression of a constitutively active form of Rab5 GTPase (Rab5-CA), which accelerates the fusion of endocytic vesicles with early endosomes, or a dominant negative form of Rab7 (*Rab7-DN*) or Rab9 (*Rab9-DN*) GTPase, which prevents transport of proteins away from the sorting endosomes, failed to suppress the supernumerary neuroblast phenotype in *vps35* mutant brains (*Figure 5—figure supplement 2*). In

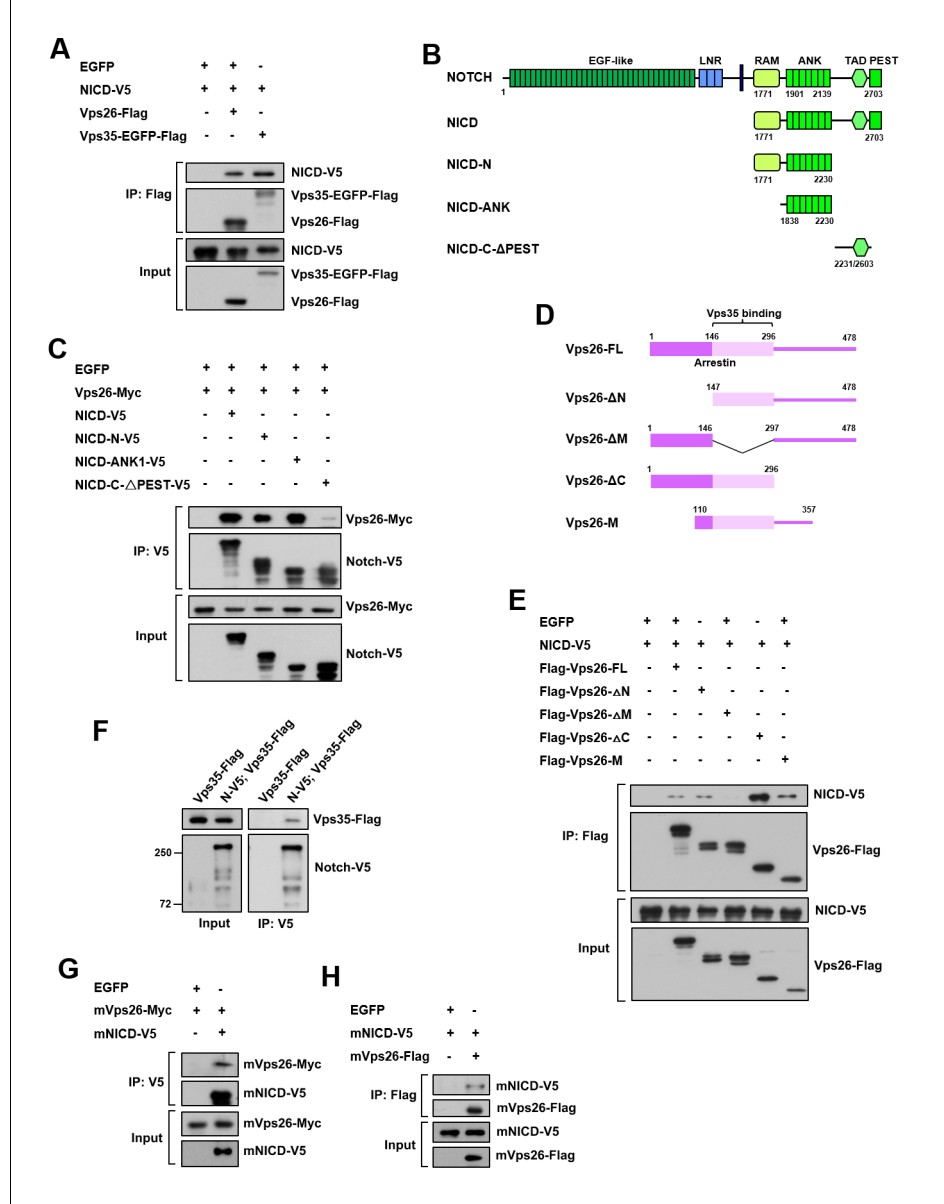

**Figure 4.** Retromer physically interacts with Notch. (**A**) Coimmunoprecipitation (CoIP) of FLAG-tagged Vps26 or Vps35 and V5-tagged Notch intracellular domain (NICD) in HEK293T cell extracts. Note that in these and subsequent panels, EGFP served as a negative control. (**B**) Schematic drawings of NICD protein domains and truncated constructs. (**C**) CoIP of full-length (FL) or truncated NICD-V5 and Vps26-Myc in HEK293T cells. (**D**) Schematic drawings of Vps26 protein domains and truncated constructs. (**E**) The reciprocal coIP of full-length (FL) or truncated FLAG-Vps26 and NICD-V5 in HEK293T cells. (**F**) CoIP of Vps35-FLAG and Notch-V5 (N–V5) in fly larval brain extracts. Note that Vps35-FLAG and N-V5 were specifically expressed in neuroblast lineages by *insc*-Gal4. (**G,H**) CoIP of Myc-tagged mouse Vps26 (mVps26-Myc) and V5-tagged mouse NICD (mNICD-V5) and the reciprocal coIP of mVps26-FLAG and mNICD-V5 in HEK293T cell extracts.
DOI: https://doi.org/10.7554/eLife.38181.015

addition, the Delta ligand clearly colocalized with Rab7⁺ enlarged endosomes in *vps35* mutant cells (*Figure 5G*). Taken together, we concluded that the enlarged, aberrant endosomal vesicles with both early and late endosomal identities are the ligand-dependent activating compartments of the Notch receptor in *vps35* mutant neural progenitors.

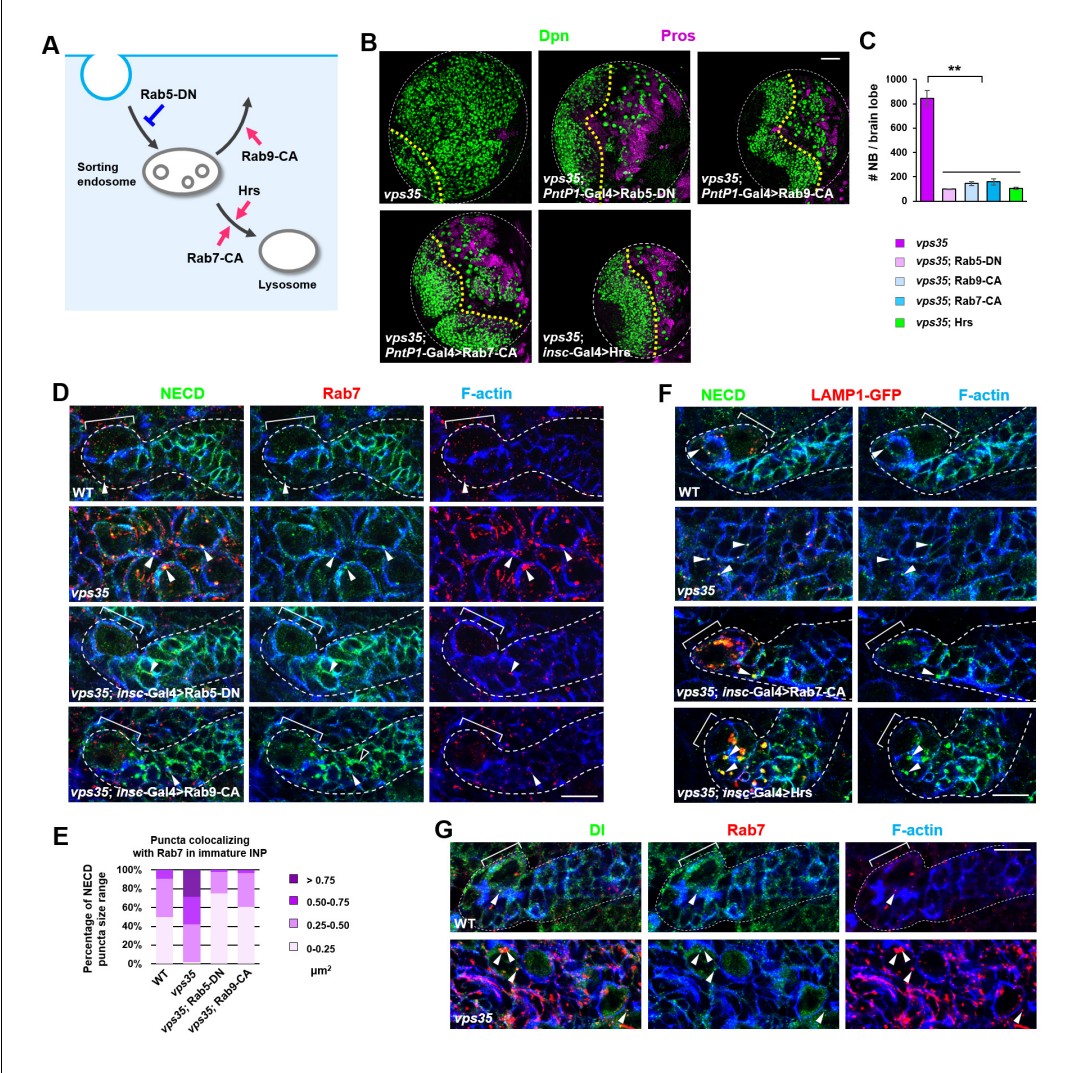

**Figure 5.** Retromer prevents intracellular ectopic cleavage of Notch receptors. (**A**) Schematic depicting a simplified endocytic pathway. Red arrow: promotion; blue flat line: inhibition. (**B,C**) Larval brain lobes of indicated genotypes were stained for Dpn and Pros. Quantification of total neuroblast number per brain lobe is shown in (**C**). **p<0.001 (n = 10–12). (**D,E**) Type II neuroblast lineages of indicated genotypes were stained for NECD, Rab7 and F-actin. NECD puncta colocalizing with Rab7 are marked with arrowheads. Quantification of the size range of Notch puncta colocalizing with Rab7[+] endosomes is shown in (**E**). (**F**) Type II neuroblast lineages of indicated genotypes were stained for NECD, GFP and F-actin. (**G**) Type II neuroblast lineages of indicated genotypes were stained for Delta, Rab7 and F-actin. Scale bars, 50 μm (**B**) and 10 μm (**D,F,G**).

DOI: https://doi.org/10.7554/eLife.38181.016

The following source data and figure supplements are available for figure 5:

**Source data 1.** Input data for bar graph *Figure 5C*.
DOI: https://doi.org/10.7554/eLife.38181.019

**Figure supplement 1.** The effects of *Rab5-DN*, Rab9-CA, Rab7-CA or Hrs expression on Notch receptor trafficking.
DOI: https://doi.org/10.7554/eLife.38181.017

**Figure supplement 2.** Overexpression of Rab5-CA, *Rab7-DN* or *Rab9-DN* failed to inhibit the brain tumor phenotype in *vps35* mutants.
DOI: https://doi.org/10.7554/eLife.38181.018

## Retromer recycles hypo-ubiquitinated Notch receptors

Why Notch needs to be transported away from its activating compartments by retromer under physiological conditions? Previous studies indicated that the internalized Notch receptors are either sorted through the ESCRT pathway and get degraded in lysosomes or recycled back to the plasma membrane for ligand binding and activation (*Kopan, 2012*). Furthermore, ubiquitin is a crucial

sorting signal for Notch receptor trafficking. We therefore considered the intriguing possibility that a pool of hypo-ubiquitinated Notch receptors might not be sorted through ESCRT-0 but instead trapped at the limiting membrane of MVBs, where they are retrieved and transported away by retromer in a timely manner.

If this hypothesis is correct, one would expect that an elevation in the activity of the E3 ubiquitin ligase(s) that promotes Notch polyubiquitination and lysosomal degradation may reduce the pool of hypo-ubiquitinated Notch in retromer mutant neural progenitors and thereby alleviate the brain tumor phenotype. Neuroblast lineage-specific overexpression of HECT domain E3 ubiquitin ligase Itch/Su(dx) or Nedd4, known for mediating Notch receptor polyubiquitination and degradation (*Cornell et al., 1999*; *Le Bras et al., 2011*; *Qiu et al., 2000*; *Sakata et al., 2004*; *Wilkin et al., 2004*), showed little inhibitory effect on the supernumerary neuroblast phenotype in *vps35* mutants (*Figure 6A,B*), suggesting that these two E3 ligases are not fully active upon overexpression in neuroblast lineages. Since Ndfip protein (Nedd4 family interacting protein) has been reported to recruit and activate Itch/Su(dx) or Nedd4 by relieving their autoinhibition caused by intramolecular interaction (*Dalton et al., 2011*; *Mund and Pelham, 2009*), we coexpressed Ndfip in an attempt to boost the catalytic activity of Itch/Su(dx) and Nedd4. Whereas simultaneous overexpression of Nedd4 and Ndfip barely exhibited any effect on brain tumor phenotype caused by *vps35* mutation (*Figure 6—figure supplement 1A–C*), coexpression of Su(dx) and Ndfip indeed led to a complete rescue of the supernumerary neuroblast phenotype in *vps35* mutants (*Figure 6A,B*). Consistent with these observations, the high accumulation of Notch in aberrant endosomal vesicles in *vps35* mutant cells was also effectively suppressed by Su(dx) and Ndfip coexpression (*Figure 6—figure supplement 1D,E*).

A related and more important prediction of this hypothesis is that the activity of the E3 ubiquitin ligase(s) targeting Notch for polyubiquitination and degradation is inherently inefficient in fly neuroblast lineages and depends on retromer-mediated retrieval to avoid ectopic accumulation and processing of Notch in INPs. If this model is correct, we reason that a reduction in the activity of the E3 ubiquitin ligase(s) might tilt the balance and lead to a larger pool of hypo-ubiquitinated Notch than normal. If retromer is meanwhile not fully functional, Notch receptors may be stalled in MVBs and eventually result in progenitor-derived tumor. Indeed, we observed a strong synergistic interaction between Su(dx) and Vps26 in mediating neuroblast self-renewal. While expression of either *vps26-RNAi* or Su(dx)-C917A, a dominant negative form of Su(dx) (*Su(dx)-DN*) that lacks its E3 ubiquitin ligase activity (*Wang et al., 2015*), by *PntP1*-Gal4, led to a mild neuroblast overproliferation phenotype (*Figure 6C,D*), simultaneous expression of *vps26-RNAi* and *Su(dx)-DN* resulted in a severe brain tumor phenotype (*Figure 6C,D*). More significantly, Notch receptors were highly accumulated in enlarged Rab7-positive endosomal vesicles in neural progenitors expressing both *vps26-RNAi* and *Su(dx)-DN*, but not in neural progenitors expressing either *vps26-RNAi* or *Su(dx)-DN* alone (*Figure 6E,F*). Immunostaining with our newly-raised Su(dx) and Ndfip antibodies (*Figure 6—figure supplement 2A,B*) revealed that Su(dx) mainly localized to the cell cortex, whereas Ndfip primarily distributed in intracellular vesicles (*Figure 6—figure supplement 2C,D*). Such largely distinct distribution pattern of Su(dx) and Ndfip in INPs might partially explain why Notch polyubiquitination and lysosomal degradation is inherently inefficient in neural progenitors.

A third prediction of this hypothesis is that ectopic processing and activation of the Notch receptor in *vps35* mutants are independent of the ESCRT pathway. Indeed, blocking the entry to the ESCRT pathway via depletion of Hrs, a key subunit of ESCRT-0, exhibited no effects on the brain tumor phenotypes in *vps35* mutants (*Figure 6G,H*). Taken together, these findings indicated that retromer prevents neural progenitor dedifferentiation through compensating the insufficient dampening of Notch signaling mediated by the Su(dx)/Ndfip-ESCRT-lysosomal pathway.

To further confirm this hypothesis, we assessed the cleavage status of the Notch receptor. Our model predicts that, upon retromer dysfunction, the chances for Notch to be transported back to the plasma membrane to access its E3 ubiquitin ligase(s) and obtain additional ubiquitin moieties become smaller. As a consequence, a pool of hypo-ubiquitinated Notch might accumulate and ectopically processed. In accordance, our results clearly showed that, a smear of presumably ubiquitinated NICD migrating at approximately 130 kilodaltons (kDa) and a un-ubiquitinated NICD band migrating at 100 kDa specifically accumulated in *vps35* mutant but not wild type brain extracts, indicating that these Notch fragments are hypo-ubiquitinated NICD (*Figure 6I*). Importantly, such increased intensity of the smear of these hypo-ubiquitinated Notch fragments in *vps35* mutants (green arrowhead) was essentially reduced back to normal upon coexpression of Su(dx) and Ndfip

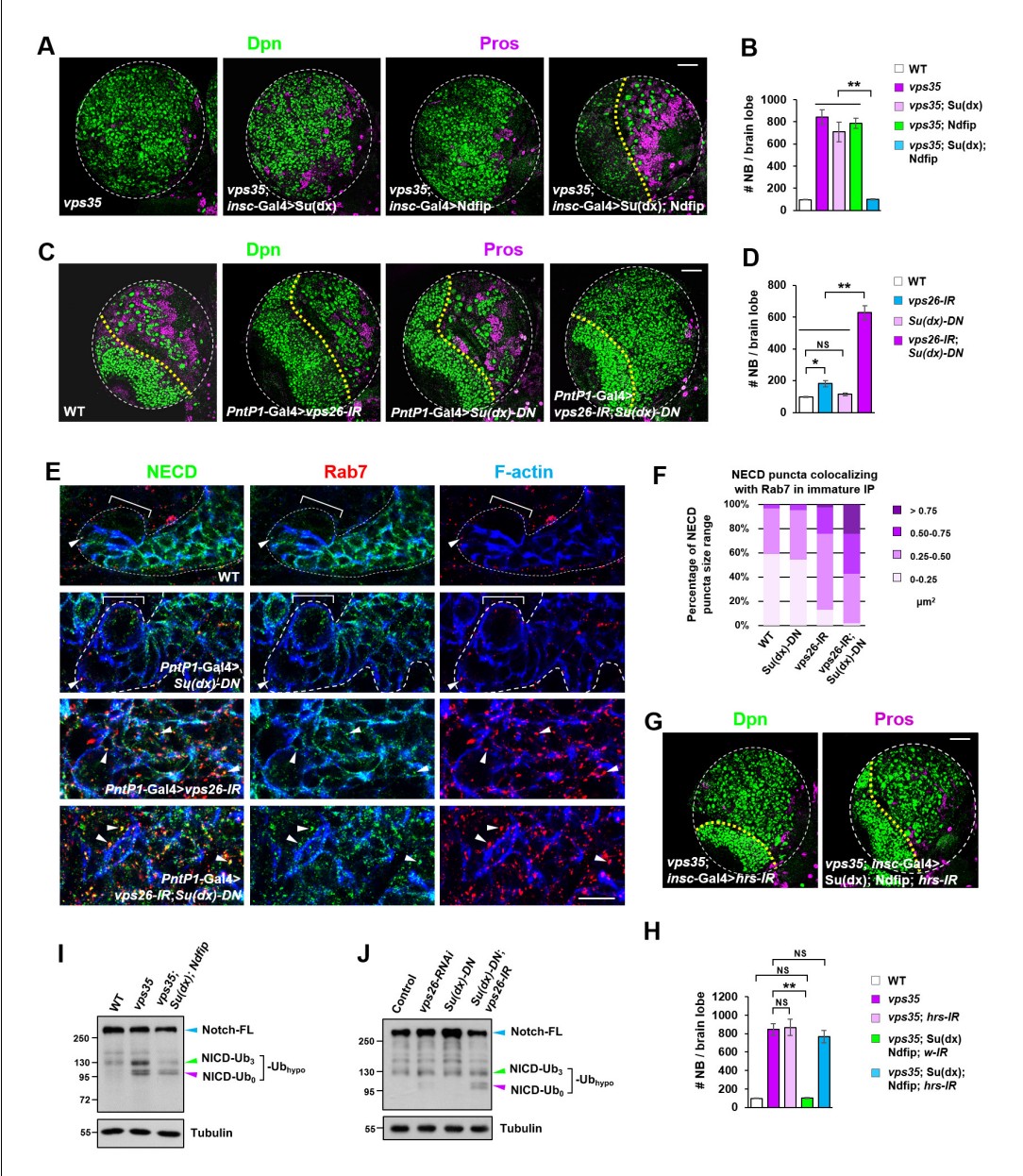

**Figure 6.** Retromer compensates for the inefficient Notch polyubiquitination and lysosomal degradation. (**A,B**) Neuroblast-specific coexpression of both Su(dx) and Ndfip but not either alone potently inhibited brain tumor phenotype in *vps35* mutants. Quantification of total neuroblast number of indicated genotypes is shown in (**B**). **p<0.001 (n = 12–15). (**C,D**) Larval brain lobes of indicated genotypes were stained for Dpn and Pros. Quantification of total neuroblast number per brain lobe is shown in (**D**). **p<0.001; *p<0.01; NS, not significant (n = 13–15). (**E**) Type II neuroblast lineages of indicated genotypes were stained for NECD, Rab7 and F-actin. NECD puncta colocalizing with Rab7 are marked with arrowheads. (**F**) Quantification of the size range of Notch puncta colocalizing with Rab7+ endosomes in immature or dedifferentiating neural progenitors of indicated genotypes. (**G,H**) Larval brain lobes of indicated genotypes were stained for Dpn and Pros. Quantification of total neuroblast number per brain lobe is shown in (**H**). **p<0.001; NS, not significant (n = 11–15). (**I,J**) Western blot analysis of larval brain extracts of indicated genotypes using anti-NICD antibody. Anti-α-tubulin blot served as a loading control. Note that hypo-ubiquitinated NICD fragments included NICD carrying approximately three ubiquitin moieties (NICD-Ub$_3$) and un-ubiquitinated NICD (NICD-Ub$_0$). Scale bars, 10 μm (**E**) and 50 μm (**A,C,G**).
DOI: https://doi.org/10.7554/eLife.38181.020

The following source data and figure supplements are available for figure 6:

**Source data 1.** Input data for bar graph *Figure 6B,D,H*.
DOI: https://doi.org/10.7554/eLife.38181.025

**Figure supplement 1.** Coexpression of Su(dx) and Ndfip specifically suppresses brain tumor phenotypes in *vps35* mutants.

*Figure 6 continued on next page*

*Figure 6 continued*

DOI: https://doi.org/10.7554/eLife.38181.021

**Figure supplement 2.** Distribution patterns of Su(dx) and Ndfip in type II neuroblast lineages.

DOI: https://doi.org/10.7554/eLife.38181.022

**Figure supplement 3.** Proteolytic processing of NiGFP in *Notch* mutant or *Notch; vps35* double mutant larval brain extracts.

DOI: https://doi.org/10.7554/eLife.38181.023

**Figure supplement 4.** Co-overexpression of Su(dx) and Ndfip caused loss of type II neuroblast lineages and brain tissue atrophy.

DOI: https://doi.org/10.7554/eLife.38181.024

(*Figure 6I*). We reason that coexpression of E3 ligase led to a reduction in the pool of hypo-ubiquitinated Notch and a corresponding increase in the pool of Notch harboring sufficient ubiquitin moieties, which was sorted through the ESCRT pathway and degraded in lysosomes. Indeed, blocking the cargo entry into the ESCRT pathway by Hrs downregulation potently inhibited the rescue effects of Su(dx)/Ndfip on retromer inactivation-induced brain tumor phenotype (*Figure 6G,H*). Consistently, un-ubiquitinated NICD fragments also specifically accumulated in larval brain extracts coexpressing *vps26-RNAi* and *Su(dx)-DN* but not in extracts expressing either *vps26-RNAi* or *Su(dx)-DN* alone (*Figure 6J*). Furthermore, in *N*; NiGFP background, in which a bacterial artificial chromosome (BAC) transgene expressing a GFP-tagged Notch (NiGFP) functionally replaces endogenous Notch (*Couturier et al., 2012*), accumulation of hypo-ubiquitinated NICD-GFP fragments was also specifically detected in *vps35* mutant but not wild type control brain extracts (*Figure 6—figure supplement 3*).

Collectively, our results supports a safeguard model whereby Notch polyubiquitination mediated by the E3 ubiquitin ligase Itch/Su(dx) is inherently inefficient within neural progenitors, relying on retromer-mediated retrograde trafficking to retrieve the pool of hypo-ubiquitinated Notch that fails to enter the ESCRT-lysosomal degradation pathway in a timely manner (*Figure 7*). Upon retromer inactivation, hypo-ubiquitinated Notch accumulates in MVBs, ectopically processed in a ligand-dependent fashion, leading to cell-autonomous activation of Notch signaling, neural progenitor dedifferentiation and tumorigenesis (*Figure 7*).

## Discussion

Unidirectional Notch signaling is a widely used strategy for initiating and maintaining binary cell fates. However, the molecular mechanisms establishing the unidirectionality of Notch signaling in stem cell lineages remain unclear. Here we reveal that, while asymmetric partition of Numb leads to a biased internalization of the Notch receptor and hence asymmetric dampening of Notch signaling in neural progenitors, it meanwhile poses a high risk of non-canonical endosomal activation of Notch. We find that the retromer complex is the key protein trafficking machinery that resolves this crisis through a timely retrieval of the Notch receptor from its endosomal activation compartments. Upon retromer dysfunction, neural progenitors dedifferentiate into neural stem cell-like status and result in the formation of transplantable tumors. Therefore, retromer acts as a tumor suppressor in *Drosophila* larval brains. Importantly, mammalian Vps35 physically interacts with Notch, colocalizes with Notch in neural progenitors, and its neuroblast-lineage-specific expression fully rescues neural progenitor-derived brain tumor phenotype in *vps35* mutants. Thus, the brain tumor suppressor function of retromer is likely to be conserved in mammals. Intriguingly, downregulation of the retromer complex components has been reported in various human cancers, including glioblastoma (*An et al., 2012*; *Bredel et al., 2005*; *Lee et al., 2006b*). Our studies thus provide a new mechanistic link between the retromer complex and carcinogenesis.

Why the E3 ubiquitin ligase system promoting Notch receptor polyubiquitination and degradation is inherently inefficient in neuroblast lineages? We speculate that Notch is probably not the only substrate of Su(dx) and Ndfip in neuroblasts or neural progenitors. Therefore, high levels and/or activity of this E3 ubiquitin ligase system above certain threshold may potentially cause imbalanced homeostasis of its critical substrates and hence perturbed neuroblast lineages. Indeed, co-overexpression of Su(dx) and Ndfip led to drastically reduced number of neuroblast lineages and severe tissue atrophy (*Figure 6—figure supplement 4*). In this case, a relatively general yet inefficient ubiquitination-degradation system coupled with a highly efficient and selective cargo retrieving

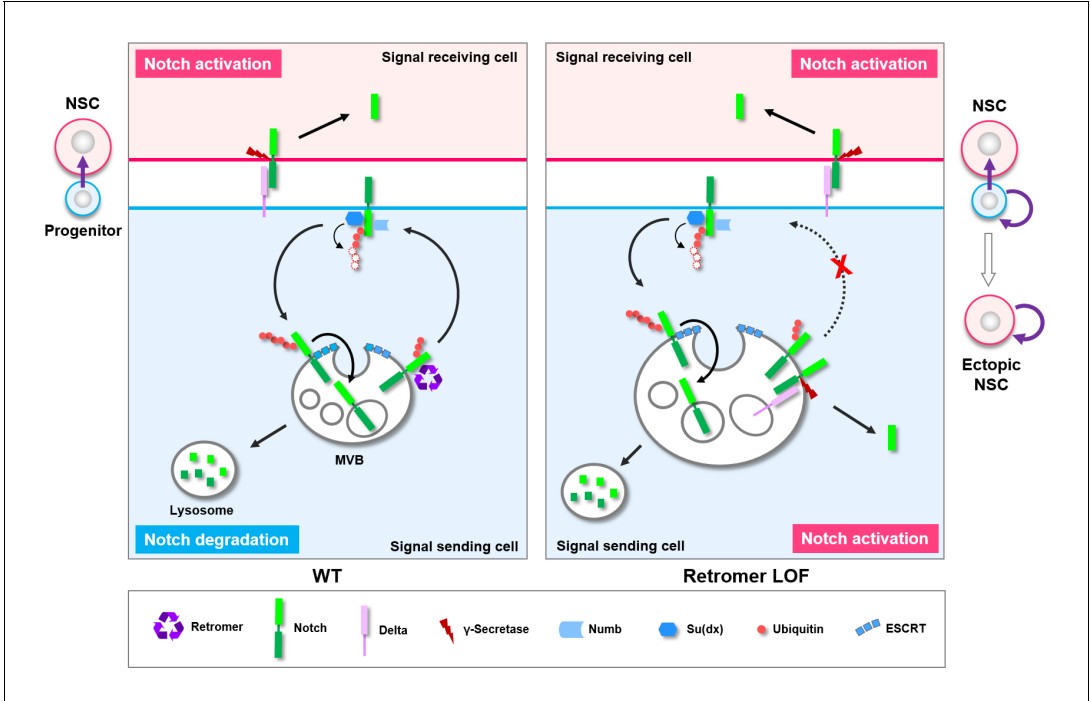

**Figure 7.** Working model. A graphic model depicting a safeguard mechanism whereby retromer ensures unidirectional Notch signaling (purple arrow) from neural progenitor (light blue) to neural stem cell (NSC; pink) by preventing cell-autonomous ectopic Notch signaling activation in neural progenitors. Retromer (purple) normally interacts with Notch (green) and retrieves the pool of hypoubiquitinated Notch evading the ESCRT (cyan)-lysosomal degradation pathway and sends it back to the cell surface (left panel). Upon retromer dysfunction, hypoubiquitinated Notch is accumulate in MVBs and aberrantly cleaved by γ-secretase (brown) in a ligand (light purple)-dependent manner, causing neural progenitor-originated tumorigenesis (right panel).

DOI: https://doi.org/10.7554/eLife.38181.026

system provides a customized regulation of the Notch receptor, ensuring sufficient dampening of Notch signaling in neural progenitors without devastating side effects.

Intriguingly, previous studies posited that retromer dysfunction causes increased levels of APP (β-amyloid precursor protein) to reside in the endosomes for longer duration than normal, resulting in accelerated processing of APP into amyloid-β, a neurotoxic fragment implicated AD pathogenesis (*Small and Gandy, 2006*; *Small and Petsko, 2015*). Furthermore, retromer maintains the integrity of photoreceptors by avoiding persistent accumulation of rhodopsin in endolysosomal compartments that stresses photoreceptors and causes their degeneration (*Wang et al., 2014*). Taken together with our study here, these findings indicate that retromer serves as bomb squad to retrieve and dis-arm harmful or toxic protein fragments from endosomes in a timely manner and thereby safeguard the integrity and fitness of the neuronal lineages.

How is the Notch receptor ectopically activated in retromer mutants? We favor the idea that Notch is activated in MVBs in a ligand-dependent, cell-autonomous manner, distinct from the major-ity of non-canonical Notch activation mechanisms. Most of the endosomal Notch activation events identified before, including ectopic Notch signaling activation in ESCRT mutants, BLOS2 mutants, or Rme8 and Vps26 double knockdown background, as well as Hif-alpha-dependent activation of Notch signaling implicated in crystal cell maintenance and survival, are all ligand-independent (*Baron, 2012*; *Childress et al., 2006*; *Gallagher and Knoblich, 2006*; *Giebel and Wodarz, 2006*; *Gomez-Lamarca et al., 2015*; *Hori et al., 2011*; *Jaekel and Klein, 2006*; *Mukherjee et al., 2011*; *Palmer and Deng, 2015*; *Schneider et al., 2013*; *Thompson et al., 2005*; *Vaccari and Bilder, 2005*; *Vaccari et al., 2009*; *Wilkin et al., 2008*; *Zhou et al., 2016*). It has been proposed that the proteases within the acidifying environment of MVB lumen are sufficient to remove the extracellular domain of Notch, leading to the S3 cleavage of Notch at the limiting membrane (*Palmer and Deng, 2015*; *Wilkin et al., 2008*). Strongly supporting this notion, blocking the entry of Notch into the

ESCRT pathway but not ligand inactivation potently inhibited ectopic Notch activation induced by ESCRT mutations (*Childress et al., 2006*; *Gallagher and Knoblich, 2006*; *Jaekel and Klein, 2006*). In sharp contrast to these previously-revealed mechanisms, attenuating ligand activity but not preventing Notch from entering the ESCRT pathway effectively rescues Notch overactivation phenotype caused by retromer dysfunction (*Figures 3D,E* and *6G,H*). Then how Notch signaling is ectopically activated in a ligand-dependent manner in retromer mutants? We speculate that, upon retromer dysfunction, both Notch and Delta are entrapped in MVBs, where Notch and Delta are presented by limiting membrane and intravesicular membrane respectively and result in ligand-dependent Notch processing and activation, resembling the scenario presented for ligand-dependent Notch signaling activation in Sara endosome (*Coumailleau et al., 2009*; *Kressmann et al., 2015*). The detailed regulatory mechanisms underlying Notch overactivation in retromer mutants warrants future investigation.

The ability of *vps35* mutant neoplastic neuroblasts to metastasize upon transplantation is intriguing (*Figure 1J,K*). Metastasis of brain tumor cells derived from neuroblast lineages has never been observed in the developing fly larval brains, likely because the limited time window of fly larval development precludes tumor progression and metastasis. Transplantation assay (*Rossi and Gonzalez, 2015*), however, provides the ectopic microenvironment and allows cancer progression in a much longer time scale (months, or even years upon retransplantation). Importantly, mutations that caused metastasis of fly brain tumor cells upon transplantation have also been implicated in various human cancers (*Caussinus and Gonzalez, 2005*; *Eroglu et al., 2014*; *Froldi et al., 2015*; *Knoblich, 2010*; *Landskron et al., 2018*; *Liu et al., 2017*; *Narbonne-Reveau et al., 2016*). Future studies on the transcriptional profiling of the distal metastatic colonies and stepwise characterization of this long-range metastatic process promise to provide us with fresh mechanistic insights into the enormously complex process of cancer metastasis.

# Materials and methods

## Key resources table

| Reagent type (species) or resource | Designation | Source or reference | Identifiers | Additional information |
|---|---|---|---|---|
| Genetic reagent (*D. melanogaster*) | *insc*-Gal4 | (*Luo et al., 1994*) | N/A | |
| Genetic reagent (*D. melanogaster*) | *PntP1*-Gal4 | (*Zhu et al., 2011*) | N/A | |
| Genetic reagent (*D. melanogaster*) | *ase*-Gal4 | (*Zhu et al., 2006*) | N/A | |
| Genetic reagent (*D. melanogaster*) | *erm*-Gal4 (II) | (*Xiao et al., 2012*) | N/A | |
| Genetic reagent (*D. melanogaster*) | *erm*-Gal4 (III) | (*Pfeiffer et al., 2008*; *Weng et al., 2010*) | N/A | |
| Genetic reagent (*D. melanogaster*) | UAS-*vps26*-RNAi | Bloomington Drosophila Stock Center | RRID: BDSC_38937 | |
| Genetic reagent (*D. melanogaster*) | UAS-*Notch*-RNAi | Vienna Drosophila RNAi Center | RRID: VDRC_27229 | |
| Genetic reagent (*D. melanogaster*) | UAS-*Delta*-RNAi | Bloomington Drosophila Stock Center | RRID: BDSC_34322 | |
| Genetic reagent (*D. melanogaster*) | UAS-*wg*-RNAi | Bloomington Drosophila Stock Center | RRID: BDSC_32994 | |
| Genetic reagent (*D. melanogaster*) | UAS-*med*-RNAi | Bloomington Drosophila Stock Center | RRID: BDSC_52214 | |
| Genetic reagent (*D. melanogaster*) | UAS-*Hrs*-RNAi | Bloomington Drosophila Stock Center | RRID: BDSC_34086; BDSC_33900 | |
| Genetic reagent (*D. melanogaster*) | UAS-*white*-RNAi | Bloomington Drosophila Stock Center | RRID: BDSC_33644 | |

*Continued on next page*

Continued

| Reagent type (species) or resource | Designation | Source or reference | Identifiers | Additional information |
|---|---|---|---|---|
| Genetic reagent (*D. melanogaster*) | UAS-Vps26-Myc | This paper | N/A | |
| Genetic reagent (*D. melanogaster*) | UAS-Vps35-FLAG | This paper | N/A | |
| Genetic reagent (*D. melanogaster*) | UAS-hVps35-FLAG | This paper | N/A | |
| Genetic reagent (*D. melanogaster*) | UAS-HA-Rab9-CA | This paper | N/A | |
| Genetic reagent (*D. melanogaster*) | UAS-Myc-Su(dx) | This paper | N/A | |
| Genetic reagent (*D. melanogaster*) | UAS-Myc-Su(dx)-C917A | This paper | N/A | |
| Genetic reagent (*D. melanogaster*) | UAS-Myc-Nedd4 | This paper | N/A | |
| Genetic reagent (*D. melanogaster*) | UAS-FLAG-Ndfip | This paper | N/A | |
| Genetic reagent (*D. melanogaster*) | UAS-Su(dx) | Bloomington Drosophila Stock Center | RRID: BDSC_51664 | |
| Genetic reagent (*D. melanogaster*) | UAS-Hrs | (*Lloyd et al., 2002*) | N/A | |
| Genetic reagent (*D. melanogaster*) | UAS-Spdo-GFP | (*Song and Lu, 2012*) | N/A | |
| Genetic reagent (*D. melanogaster*) | UAS-*Dl-DN* | Bloomington Drosophila Stock Center | RRID: BDSC_26698 | |
| Genetic reagent (*D. melanogaster*) | UAS-*Psn-DN* | Bloomington Drosophila Stock Center | RRID: BDSC_8323 | |
| Genetic reagent (*D. melanogaster*) | UAS-*Kuz-DN* | Bloomington Drosophila Stock Center | RRID: BDSC_6578 | |
| Genetic reagent (*D. melanogaster*) | UAS-*EGFR-DN* | Bloomington Drosophila Stock Center | RRID: BDSC_5364 | |
| Genetic reagent (*D. melanogaster*) | UAS-*ptc-DN* | Bloomington Drosophila Stock Center | RRID: BDSC_31928 | |
| Genetic reagent (*D. melanogaster*) | UAS-GFP-LAMP1 | Bloomington Drosophila Stock Center | RRID: BDSC_42714 | |
| Genetic reagent (*D. melanogaster*) | UAS-mito-GFP | Bloomington Drosophila Stock Center | RRID: BDSC_8442 | |
| Genetic reagent (*D. melanogaster*) | sqh-EYFP-Golgi | Bloomington Drosophila Stock Center | RRID: BDSC_7193 | |
| Genetic reagent (*D. melanogaster*) | UAS-YFP-Rab5-WT | Bloomington Drosophila Stock Center | RRID: BDSC_24616 | |
| Genetic reagent (*D. melanogaster*) | UAS-YFP-Rab5-CA | Bloomington Drosophila Stock Center | RRID: BDSC_9773 | |
| Genetic reagent (*D. melanogaster*) | UAS-*Rab5-DN* | Bloomington Drosophila Stock Center | RRID: BDSC_42704 | |
| Genetic reagent (*D. melanogaster*) | UASp-*YFP-Rab7-DN* | Bloomington Drosophila Stock Center | RRID: BDSC_9778 | |
| Genetic reagent (*D. melanogaster*) | UAS-Rab7-CA | Bloomington Drosophila Stock Center | RRID: BDSC_42707 | |
| Genetic reagent (*D. melanogaster*) | UASp-YFP-Rab9-WT | Bloomington Drosophila Stock Center | RRID: BDSC_9784 | |
| Genetic reagent (*D. melanogaster*) | UASp-YFP-Rab9-CA | Bloomington Drosophila Stock Center | RRID: BDSC_9785 | |

*Continued on next page*

*Continued*

| Reagent type (species) or resource | Designation | Source or reference | Identifiers | Additional information |
|---|---|---|---|---|
| Genetic reagent (*D. melanogaster*) | UASp-*YFP-Rab9-DN* | Bloomington Drosophila Stock Center | RRID: BDSC_23642 | |
| Genetic reagent (*D. melanogaster*) | UAS-*YFP-Rab11-DN* | Bloomington Drosophila Stock Center | RRID: BDSC_9792 | |
| Genetic reagent (*D. melanogaster*) | *vps35*$^{E42}$ | Gift from Xinhua Lin (**Belenkaya et al., 2008**) | N/A | |
| Genetic reagent (*D. melanogaster*) | *vps35*$^1$ | Gift from Xinhua Lin (**Belenkaya et al., 2008**) | N/A | |
| Genetic reagent (*D. melanogaster*) | $N^{55e11}$; NiGFP | (**Couturier et al., 2012**) | N/A | |
| Genetic reagent (*D. melanogaster*) | UAS-FLP, Ubi-p63E-FRT-nlsGFP | Bloomington Drosophila Stock Center | RRID: BDSC_28282 | |
| Antibody | Mouse anti-Notch$^{ECD}$ (C458.2H) | Developmental Studies Hybridoma Bank | RRID: AB_528408 | IHC (1:80) |
| Antibody | Mouse anti-Pros (MR1A) | Developmental Studies Hybridoma Bank | RRID: AB_528440 | IHC (1:100) |
| Antibody | Rat anti-Mira | Abcam | Cat#Ab197788 | IHC (1:100) |
| Antibody | Rabbit anti-Dpn | Gift from Y.N. Jan | N/A | IHC (1:1000) |
| Antibody | Guinea pig anti-Numb | Gift from J. Skeath (**O'Connor-Giles and Skeath, 2003**) | N/A | IHC (1:1000) |
| Antibody | Mouse anti-β-galactosidase (40-1a) | Developmental Studies Hybridoma Bank | RRID: AB_2314509 | IHC (1:100) |
| Antibody | Guinea pig anti-Ase | Gift from Y.N. Jan | N/A | IHC (1:400) |
| Antibody | Rabbit anti-aPKC ζ C20 | Santa Cruz Biotechnologies | RRID: AB_2168668 | IHC (1:1000) |
| Antibody | Rabbit anti-Rab7 | Gift from A. Nakamura (**Tanaka and Nakamura, 2008**) | N/A | IHC (1:2000) |
| Antibody | Mouse anti-Wg (4D4) | Developmental Studies Hybridoma Bank | RRID: AB_528512 | IHC (1:100) |
| Antibody | Mouse anti-Ptc (Apa 1) | Developmental Studies Hybridoma Bank | RRID: AB_528441 | IHC (1:100) |
| Antibody | Rabbit anti-Myc (71D10) | Cell Signaling Technology | RRID: AB_10693332 | WB (1:2000) |
| Antibody | Rabbit anti-FLAG | Sigma-Aldrich | RRID: AB_439687 | IHC (1:1000) |
| Antibody | Rabbit anti-V5 | Sigma-Aldrich | RRID: AB_261889 | WB (1:1000) |
| Antibody | Mouse anti-c-Myc | CW Biotech | Cat#cw0299M | WB (1:2000) |
| Antibody | Mouse anti-Delta$^{ECD}$ (C594.9B) | Developmental Studies Hybridoma Bank | RRID: AB_528194 | IHC (1:200) |
| Antibody | Mouse anti-Notch$^{ICD}$ (C17.9C6) | Developmental Studies Hybridoma Bank | RRID: AB_528410 | WB (1:1000) |
| Antibody | Rabbit anti-Vps26 | This paper | N/A | IHC (1:200) |
| Antibody | Rabbit anti-Su(dx) | This paper | N/A | IHC (1:200) |
| Antibody | Rabbit anti-Ndfip | This paper | N/A | IHC (1:100) |
| Antibody | Anti-V5 affinity gels | Sigma-Aldrich | RRID: AB_10062721 | 15 μl gel per coIP reaction |
| Antibody | Anti-FLAG M2 affinity gels | Sigma-Aldrich | RRID: AB_10063035 | 15 μl gel per coIP reaction |
| Software, algorithm | ImageJ | NIH | N/A | |
| Software, algorithm | Photoshop CS5 | Adobe | N/A | |
| Software, algorithm | The Leica Application Suite 2.6.3 | Leica | N/A | |
| Cell line (Human) | HEK293T | ATCC | RRID: CRL-3216 | |
| Recombinant DNA reagent | pcDNA3.1 | Invitrogen | Cat#: V79020 | |

*Continued on next page*

Continued

| Reagent type (species) or resource | Designation | Source or reference | Identifiers | Additional information |
|---|---|---|---|---|
| Recombinant DNA reagent | vps26 (*Drosophila* cDNA) | BDGP | LD29140 | |
| Recombinant DNA reagent | vps35-RB (*Drosophila* cDNA) | BDGP | SD03023 | |
| Recombinant DNA reagent | nedd4-RK (*Drosophila* cDNA) | BDGP | SD04682 | |
| Recombinant DNA reagent | vps35 (*human* cDNA) | Human ORFeome | Internal ID: 7965 Genbank Accession: CV029249 | |
| Recombinant DNA reagent | vps26A (*human* cDNA) | Addgene | Cat#17636 | |
| Recombinant DNA reagent | NICD1 (*mouse* cDNA) | Addgene | Cat#20183 | |

## Fly genetics

Fly culture and crosses were performed according to standard procedures. *Drosophila* stocks used in this study include: *vps35$^{E42}$* (*Belenkaya et al., 2008*)(a gift from Dr. Xinhua Lin); *vps35$^1$* (*Belenkaya et al., 2008*); *vps26$^{G2008}$* (BL26623); UAS-Vps35-FLAG (this study); UAS-Vps26-Myc (RR: RNAi resistant form; this study); UAS-*vps26-RNAi* (BL38937); UAS-*Notch-RNAi* (VDRC27229); UAS-*Dl-DN* (BL26698); UAS-*Dl-RNAi* (BL34322); UAS-*Psn-DN* (BL8323); UAS-*Kuz-DN* (BL6578); UAS-*Rab5-DN* (BL42704); UAS-Rab7-CA (BL42707); UAS-Hrs (*Lloyd et al., 2002*); UAS-GFP-LAMP1 (BL42714); UAS-mito-GFP (BL8442); UASp-YFP-Rab9-WT (BL9784); UASp-YFP-Rab9-CA (BL9785); UAS-Su(dx) (BL51664); UAS-Myc-Su(dx) (this study); UAS-FLAG-Ndfip (this study); UAS-Myc-Su(dx)-C917A (this study); UAS-Myc-Nedd4 (this study); *insc*-Gal4 (*Luo et al., 1994*); *PntP1*-Gal4 (*Zhu et al., 2011*); *ase*-Gal4 (*Zhu et al., 2006*); *erm*-Gal4 (II) (*Xiao et al., 2012*); *erm*-Gal4 (III) (*Pfeiffer et al., 2008*; *Weng et al., 2010*); E(spl)mγ-GFP (*Almeida and Bray, 2005*; *Monastirioti et al., 2010*); UAS-Spdo-GFP (*Song and Lu, 2012*); UAS-*wg-RNAi* (BL32994); UAS-*EGFR-DN* (BL5364); UAS-*med-RNAi* (BL52214); UAS-*ptc-DN* (BL31928); UAS-*Hrs-RNAi* (BL34086, BL33900); *N$^{55e11}$*; NiGFP (*Couturier et al., 2012*) and UAS-FLP, Ubi-p63E-FRT > stop > FRT-nlsGFP (BL28282) (*Evans et al., 2009*).

All larval brains phenotypes were analyzed at late third instar larval stage. Note that, compared to wild type control, the development of *vps35$^{E42}$* mutant larvae was delayed. Experiments with no special notification were carried out as follows: Eggs were collected for 4–6 hr at 25°C and kept at 25°C until dissection at late third instar larval stage.

The experimental conditions shown in *Figures 1D, F*, *3D*, *5B, D, F*, *6C, E and I* are as follows: Eggs were collected for 4–6 hr at 22°C, kept at 22°C for 24 hr (*Figures 1D, F*, *3D*, *5B*, *6C and E*) or 48 hr (*Figures 5D, F* and *6I*) after hatching and shifted to 29°C until dissection at late third instar larval stage. The experimental conditions shown in 3C is as follows: Eggs were collected for 4–6 hr at 25°C, kept at 18°C for 8 days, then shifted to 29°C for 40 hr before dissection. The experimental conditions shown in *Figure 1—figure supplement 2* and *Figure 6—figure supplement 4* are as follows: Eggs were collected for 4–6 hr at 22°C. Larvae were raised at 29°C immediately after hatching until dissection at late third instar larval stage.

## Molecular biology

Full-length cDNA clones for vps35, vps26 (LD29140), and nedd4 were obtained from Drosophila Genomics Resource Center (DGRC). For ndfip and su(dx) cDNAs, their respective coding exons were cloned by genomic DNA PCR from w$^{1118}$ flies and UAS-Su(dx) transgenic flies respectively, assembled together by the Gibson Assembly method and fully sequenced. FLAG-Ndfip, Myc-Nedd4 and Myc-Su(dx)-WT were constructed by adding a FLAG tag (DYKDDDDK) or a Myc tag (EQKLISEEDL) respectively to the N-terminus. Vps35-FLAG and Vps26-Myc were constructed by adding a FLAG tag or a Myc tag respectively to the C-terminus. Note that shmiRNA-resistant sequence was introduced into Vps26 before it was cloned into the pUAST vector. A missense mutation (C917A) was introduced into Su(dx) to generate a ligase-inactivated form. NICD-V5 was generated as described

before (*Liu et al., 2017*). All transgenic plasmids were verified by DNA sequencing before germline transformation.

For coimmunoprecipitation experiments, Vps26-FLAG and Vps26-Myc were cloned into pcDNA3.1 vector respectively (Invitrogen). Vps26 truncated forms Vps26-ΔN (aa 147–478), Vps26-ΔM (aa 1–146 and aa 297–478), Vps26-ΔC (aa 1–296) and Vps26-M (aa 110–357) were cloned with a N-terminal FLAG tag into pcDNA3.1 vector respectively. Mouse NICD cDNA was obtained from Addgene, while mouse vps26 cDNA were generated by introducing I16V, V17A, E217D to human vps26 cDNA (Addgene). NICD-V5 construct was generated as described before (*Liu et al., 2017*), except that aa 1767–1770, 1832–1835, 2202–2205 and 2222–2225 were deleted to remove its nuclear localization sequence. NICD truncated versions NICD-N (aa 1771–2230) and NICD-ANK (aa 1838–2230) were cloned into the vector pcDNA3.1 with V5 tag added to C-terminus, and NICD-C-Δ PEST (aa 2231–2603) with a V5 tag inserted between aa 2571 and 2572. To generate mouse NICD-Δ NLS-V5, aa 1749–1752, 1771–1774, 1811–1814, 2146–2149 and 2167–2170 were deleted from mouse NICD (aa 1744–2531 of mouse Notch1 protein) and a V5 tag was inserted between aa 2396 and 2397, before cloned into pcDNA3.1 vector. mVps26-FLAG and mVps26-Myc were cloned into the pCMV vector respectively.

## MARCM clonal analysis

Neuroblast MARCM clones were generated as previously described (*Song and Lu, 2011*). Briefly, newly hatched larvae were heat-shocked at 37°C for 90 min and further aged at 25°C for indicated time before dissection. FRTG13, *vps35[1]* was used for neuroblast MARCM clonal analysis, as shown in *Figure 1I* and *Figure 1—figure supplement 2*, with FRTG13 alone serving as a negative control.

## Immunohistochemistry

For larval brain immunostaining, larvae were dissected in Schneider's Insect Medium (Sigma-Aldrich) and proceeded as previously described (*Liu et al., 2017*; *Song and Lu, 2011*). Briefly, larval brains were fixed with 4% paraformaldehyde in PEM buffer (100 mM PIPES at pH 6.9, 1 mM EGTA, 1 mM MgCl$_2$) for 22 min at room temperature. Brains were washed several times with PBST buffer (1 × PBS plus 0.1% Triton X-100) and were incubated with appropriate primary antibody overnight at 4°C or for 2 hr at room temperature, labeled with secondary antibodies according to standard procedures, and mounted in Vectashield (Vector Laboratories). For anti-Delta staining, larval brains were fixed with 4% paraformaldehyde/PEM buffer for 20 min at room temperature, blocked in 3% BSA/PBST for 20 min at room temperature, before being incubated with mouse anti-Delta (1:200) in 0.5% BSA/PBST for 12 hr at 4°C. After washing with PBST buffer, brains were incubated with goat anti-mouse secondary antibody (1:100) in 0.5% BSA/PBST for 2 hr at room temperature before being mounted in Vectashield.

Antibodies generated in this study were rabbit anti-Vps26 antibody [GST fusion of aa 320–478, affinity purified (Abclonal Biotech.), used at 1:200], rabbit anti-Su(dx) [GST fusion of aa 350–500, affinity-purified (Abclonal Biotech.), used at 1:200] and rabbit anti-Ndfip [GST fusion of aa 2–165, affinity-purified (Abclonal Biotech.), used at 1:100]. To eliminate any non-specific binding, all antibodies were preabsorbed before being used in immunostaining experiments. Images were obtained on a Leica TCS SP8 AOBS confocal microscope and were processed with LAS AF (Leica) and Adobe Photoshop CS5.

Other primary antibodies used for immunohistochemistry were chicken anti-GFP (1:2000, Abcam), mouse anti-Pros (1:100, Developmental Studies Hybridoma Bank [DSHB]), mouse anti-N$^{ECD}$ C458.2H (1:80, DSHB), rat anti-Miranda (1:100; Abcam), rabbit anti-Dpn (1:1000, Y.N. Jan), rabbit anti-Rab7 (1:2000, a generous gift from A. Nakamura) (*Tanaka and Nakamura, 2008*); guinea pig anti-Numb (1:1000, a generous gift from J. Skeath) (*O'Connor-Giles and Skeath, 2003*), mouse anti-β-galactosidase (1:100, DSHB), guinea pig anti-Ase (1:400, Y.N. Jan), rabbit anti-aPKC ζ C20 (1:1000, Santa Cruz Biotechnologies) and mouse anti-Dl$^{ECD}$ C594.9B (1:200, DSHB). The outline of individual, dispersed neuroblast lineages was determined by the staining pattern of general cell cortex marker F-actin or CD8-GFP/CD8-RFP and marked by white dashed line.

## Cell line and transfection

Human embryonic kidney HEK293T cells (ATCC, RRID: CRL-3216; obtained from Dr. Hong Wu's laboratory, Peking University, and authenticated by ATCC) were maintained in DMEM medium (Invitrogen) supplemented with 10% FBS at 37°C and 5% CO2. DNA transfection was performed using a standard polyethylenimine (PEI) protocol. The cell line has been tested for and confirmed to be negative for mycoplasma contamination, using short tandem repeat (STR) profiling technique.

## Coimmunoprecipitation

Coimmunoprecipitation (CoIP) assays in HEK 293 T cell extracts were performed as previously described (*Liu et al., 2017*; *Song and Lu, 2012*). Briefly, 48 hr after transfection, HEK 293 T cells were harvested, washed and resuspended in lysis buffer [50 mM Tris-HCl (pH 8.0); 120 mM NaCl; 5 mM EDTA; 1% NP-40; 10% glycerol; protease inhibitor cocktail (Sigma-Aldrich); 2 mM $Na_3VO_4$] and kept on ice for 20 min. Cell extracts were sonicated with Bioruptor Plus (Biosense) at 4°C. The cell extracts were clarified by centrifugation, and proteins immobilized by binding to anti-FLAG M2 or anti-V5 (Sigma-Aldrich) affinity gel for 4 hr or overnight at 4°C. Beads were washed and proteins recovered directly in SDS-PAGE sample buffer. Rabbit anti-FLAG (Sigma-Aldrich), rabbit anti-V5 (Sigma-Aldrich) or mouse anti-c-Myc (CWBIO) were used for Western blot analysis.

For in vivo coIP, larval brains coexpressing UAS-Vps35-FLAG and UAS-Notch-V5 by *insc*-Gal4 were used as experimental group, whereas larval brains expressing UAS-Vps35-FLAG alone by *insc*-Gal4 served as control. Approximately 350 late third instar larval brains of each genotype were dissected and collected in ice-cold 1xPBS solution. Protein samples were prepared by grinding brains in lysis buffer [50 mM Tris-HCl, 120 mM NaCl, 5 mM EDTA, 10% glycerol, 1% NP-40, protease inhibitor cocktail (Sigma-Aldrich)] with a plastic pestle. Immunoprecipitation was carried out with anti-V5 affinity gels (Sigma-Aldrich).

## Transplantation assay

GFP$^+$ larval brain pieces were transplanted into the abdomen of young female adult host flies as previously described (*Caussinus and Gonzalez, 2005*; *Liu et al., 2017*). After transplantation, host flies were transferred to fresh food every day and were observed under a fluorescent scope every two days to analyze tumor formation and metastasis.

## Transmission electron microscopy (TEM)

*Drosophila* late third instar larval brains were dissected in PBS buffer, and immediately transferred into Fixation buffer I (2% paraformaldehyde/2.5% glutaraldehyde in 0.1 M phosphate buffer, pH 7.4) for 2 hr at room temperature, and then overnight at 4°C. The samples were then fixed in the Fixation buffer II (1% tannic acid/2.5% glutaraldehyde in 0.1 M phosphate buffer, pH 7.4) for 2 hr at room temperature. After rinsing several times in phosphate buffer, the brain samples were post-fixed in 2% $OsO_4$ with 1.5% Potassium Ferrocyanide for 1 hr at room temperature and stained with 2% aqueous uranyl acetate overnight at 4°C. Following several washes in distilled water, samples were dehydrated through a graded alcohol series and subsequently embedded in Spurr's resin (SPI supplies, PA, USA). Ultra-thin sections (70 nm) were cut with a diamond knife using an ultramicrotome (UC7, Leica Microsystem) and mounted on copper grids with a single slot. Sections were stained with uranyl acetate and lead citrate, and observed under a FEI Tecnai G2 Spirit transmission electron microscope at 120 kV.

## Acknowledgements

We are grateful to Drs. Hugo Bellen, Sarah J Bray, Chris Q Doe, Yuh-Nung Jan, Renjie Jiao, Tzumin Lee, Xinhua Lin, Bingwei Lu, Liqun Luo, Fumio Matsuzaki, Akira Nakamura, Yi Rao, Gerald M Rubin, François Schweisguth, James B Skeath, Esther M Verheyen, Hong Wu, University of Iowa DSHB, VDRC, Bloomington *Drosophila* Stock Center, and the TRiP at Harvard Medical School and Tsinghua University for fly stocks and reagents. We thank Dr. Ying-Chun Hu, Yun-Chao Xie and Hong-Mei Zhang for their professional technical assistance in TEM sample preparation and image analysis at the Core Facilities of School of Life Sciences, Peking University. We thank Dr. Francois Guillemot for helpful discussion, Dr. Bingwei Lu for supporting this project while it was in its infancy, Ms. Qiuju Zhu

for technical assistance, and members of the Song lab for discussions and help. This work was supported by the National Natural Science Foundation of China [31471372 and 31771629 to YS], the Peking-Tsinghua Center for Life Sciences (to YS) and the Ministry of Education Key Laboratory of Cell Proliferation and Differentiation (to YS).

## Additional information

### Funding

| Funder | Grant reference number | Author |
|---|---|---|
| National Natural Science Foundation of China | 31471372 | Yan Song |
| Peking-Tsinghua Center for Life Sciences | | Yan Song |
| Ministry of Education Key Laboratory of Cell Proliferation and Differentiation | | Yan Song |
| National Natural Science Foundation of China | 31771629 | Yan Song |

The funders had no role in study design, data collection and interpretation, or the decision to submit the work for publication.

### Author contributions

Bo Li, Conceptualization, Data curation, Formal analysis, Supervision, Funding acquisition, Methodology, Writing—original draft, Writing—review and editing; Chouin Wong, Conceptualization, Data curation, Formal analysis, Investigation, Methodology, Writing—review and editing; Shihong Max Gao, Conceptualization, Data curation, Formal analysis, Methodology, Writing—review and editing; Rulan Zhang, Data curation, Formal analysis, Investigation, Methodology, Writing—review and editing; Rongbo Sun, Resources, Formal analysis, Investigation, Methodology, Writing—review and editing; Yulong Li, Resources, Formal analysis, Supervision, Investigation, Methodology, Writing—review and editing; Yan Song, Conceptualization, Data curation, Formal analysis, Supervision, Funding acquisition, Investigation, Methodology, Writing—original draft, Writing—review and editing

### Author ORCIDs

Yan Song http://orcid.org/0000-0002-1413-6123

### Decision letter and Author response

Decision letter https://doi.org/10.7554/eLife.38181.030
Author response https://doi.org/10.7554/eLife.38181.031

## Additional files

### Supplementary files

• Transparent reporting form
DOI: https://doi.org/10.7554/eLife.38181.027

### Data availability

All data generated or analysed during this study are included in the manuscript and supporting files.

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
