## [Decision Letter]

Thank you for submitting your article "The retromer complex safeguards against neural progenitor-derived tumorigenesis by regulating Notch receptor trafficking" for consideration by *eLife*. Your article has been reviewed by Didier Stainier as the Senior Editor, a Reviewing Editor, and three reviewers. The following individuals involved in review of your submission have agreed to reveal their identity: Shinya Yamamoto (Reviewer #1); Boris Egger (Reviewer #2); Lee Tzumin (Reviewer #3).

The reviewers have discussed the reviews with one another and the Reviewing Editor has drafted this decision to help you prepare a revised submission.

The reviewers and the reviewing editor all find the manuscript interesting and showing solid and convincing evidence for retromer promoting Notch degradation through enhancing Notch ubiquitination. Most of the experiments are well designed, well controlled and well performed. These findings are significant and shed new light on Notch signaling as well as fundamental cell biology.

Before recommending publication the following points should be addressed.

Essential revisions:

Dedifferentiation: The authors conclude that immature INPs revert back to a Type II neuroblast state when retromer complex function is impaired. To substantiate this claim could the authors use additional markers. A marker for immature INPs is Earmuff that should not be present in the ectopic Type II neuroblasts. Alternatively, the authors should perform some sort of lineage tracing experiment. Such experiment has been performed in a previous study by the corresponding author (Song and Lu, 2011, Figure 2).

Ligand dependency: We are not fully convinced that the Notch over-activation phenotype is ligand-dependent. The experiments to repress Delta/Kuz (Figure 3D,E) used insc-GAL4 and PntP1-GAL4. insc-GAL4 is a neuroblast driver, whereas PntP1-GAL4 drives expression in both neuroblast and INP. It is unclear if the observed effects resulted from repressing Delta/Kuz and therefore Notch activity in neuroblast or INP. It is possible that the rescuing results were secondary due to loss of Notch within neuroblast, which eliminated the ectopic neuroblasts (rather than cell-autonomously preventing immature INP from dedifferentiation). In addition, the authors came to this conclusion because their "dominant-negative Delta" and "dominant negative Kuzbanian" was able to suppress their Notch activation and tumorigenic phenotype. However, considering that there is data in the literature that (1) "dominant negative Delta" can inhibit ligand-independent Notch signaling events (Palmer et al., 2014) and (2) some ligand-independent Notch signaling depends on Kuzbanian (Shimizu et al., 2014), I feel there is still a possibility that the Notch activation reported here is ligand-independent. Most of the studies that have demonstrated ligand-dependence/independence in the *Drosophila* Notch signaling field have directly shown this using Delta null, Serrate null and/or Delta Serrate double null alleles. If these experiments cannot be performed in this specific situation, additional experiments to modulate the function of other proteins that are known to specifically regulate ligand-dependent Notch signaling such as Neuralized/Mindbomb1 can be used to strengthen the author's claim. Also, the authors should describe the precise nature of the dominant negative reagents of Delta and Kuzbanian used in their experiments, rather than broadly labeling them as "dominant negative". Finally, the authors propose a model in which hypo-Ubiquitinated Notch becomes activated by ligands that are present in luminal vesicles within the multi-vesicular body. However, the authors do not provide any immunostaining data that shows that ligands are indeed present in the Rab5/7 positive enlarged endosomes. This is a simple experiment that can be done relatively quickly to support their model of "ligand-dependent" Notch activation that occurs upon loss of retromer function.

ESCRTs:The authors argue that the retromer-LOF phenotype is independent of the ESCRT pathway reported previously. Then, it is curious that Hrs-GOF (Figure 5A-C) rescued the vps35 mutant phenotype, whereas the Hrs-LOF (Figure 6G) had no effect. Given the use of PntP1-GAL4 in these experiments, the successful rescue by Hrs-GOF might be also caused by ectopic expression of Hrs in the neuroblast, rather than INP. Please address this concern with more specific manipulations. In addition, the authors refer to the enlarged endosomes they observe in retromer defective neuroblast-like cells as "multi-vesicular bodies" (MVBs, subsection “*vps35* mutant dedifferentiating neural progenitors contained enlarged Rab7-positive endosomal vesicles” and throughout paper) where Notch accumulates and becomes activated. However, to claim that these Rab5/Rab7 positive structures are MVBs, the authors should show the presence of enlarged endosomes/vesicles that contain intraluminal vesicles through transmission electron microscopy or some sort of super-resolution light microscopy in retromer deficient cells. Otherwise, these structures can be enlarged endosomes of mixed identity that lack internal vesicles, hence not meeting the definition of MVBs. If these structures are not MVBs, or if it is not technically feasible to analyze them further within the allocated time frame, the structure should be rather called 'enlarged endosomes' and the authors model of ligand-dependent activation of Notch occurring on the limiting membrane of MVBs through ligands present in intraluminal vesicles should be toned down/amended.

---

## [Author Response]

The reviewers and the reviewing editor all find the manuscript interesting and showing solid and convincing evidence for retromer promoting Notch degradation through enhancing Notch ubiquitination. Most of the experiments are well designed, well controlled and well performed. These findings are significant and shed new light on Notch signaling as well as fundamental cell biology.Before recommending publication the following points should be addressed.Essential revisions:Dedifferentiation: The authors conclude that immature INPs revert back to a Type II neuroblast state when retromer complex function is impaired. To substantiate this claim could the authors use additional markers. A marker for immature INPs is Earmuff that should not be present in the ectopic Type II neuroblasts. Alternatively, the authors should perform some sort of lineage tracing experiment. Such experiment has been performed in a previous study by the corresponding author (Song and Lu, 2011, Figure 2).

We wish to thank the editors and reviewers for all the efforts they have put into reviewing and improving our manuscript. We agree with the reviewers that the conclusion on dedifferentiation of immature INPs back to type II neuroblasts upon retromer dysfunction could be further substantiated. Due to the lack of a good Earmuff antibody in hand, we chose to follow the reviewers’ insightful suggestion and performed the lineage-tracing experiment using an immature INP-specific Gal4 *erm*-Gal4 (II). As shown in new Figure 1—figure supplement 4, while wild type immature progenitors differentiated into Dpn^+^ Ase^+^ mature INPs (white arrowhead) and Dpn^-^ GMCs orneurons (cyan arrowhead), *vps35* mutant immature INPs gave rise to Dpn^+^ Ase^-^ neuroblast-like cells of various cellular sizes (yellow arrowheads). These results provided compelling and direct evidence supporting that immature INPs revert their cell fate back to type II neuroblasts when retromer function is impaired. The text added to describe this lineage-tracking experiment: Subsection “The retromer complex prevents neural progenitor dedifferentiation and tumorigenesis”.

Ligand dependency: We are not fully convinced that the Notch over-activation phenotype is ligand-dependent. The experiments to repress Delta/Kuz (Figure 3D,E) used insc-GAL4 and PntP1-GAL4. insc-GAL4 is a neuroblast driver, whereas PntP1-GAL4 drives expression in both neuroblast and INP. It is unclear if the observed effects resulted from repressing Delta/Kuz and therefore Notch activity in neuroblast or INP. It is possible that the rescuing results were secondary due to loss of Notch within neuroblast, which eliminated the ectopic neuroblasts (rather than cell-autonomously preventing immature INP from dedifferentiation).

We thank the reviewers for this important point. In order to confirm that the Notch overactivation phenotype in *vps35* mutants is ligand-depend, we have now followed the reviewers’ suggestions and used the immature INP-specific driver erm-Gal4 (II), instead of insc-Gal4 or PntP1-Gal4, for performing the rescue experiments. As shown in new Figure 3D,E, immature INP-specific expression of dominant-negative Kuzbanian (Kuz-DN) potently rescued the supernumerary NB phenotype in *vps35* mutants. These results clearly indicated that repressing Kuz in immature INPs is sufficient to cell-autonomously prevent these cells from dedifferentiation.

In addition, the authors came to this conclusion because their "dominant-negative Delta" and "dominant negative Kuzbanian" was able to suppress their Notch activation and tumorigenic phenotype. However, considering that there is data in the literature that (1) "dominant negative Delta" can inhibit ligand-independent Notch signaling events (Palmer et al., 2014) and (2) some ligand-independent Notch signaling depends on Kuzbanian (Shimizu et al., 2014), I feel there is still a possibility that the Notch activation reported here is ligand-independent. Most of the studies that have demonstrated ligand-dependence/independence in the Drosophila Notch signaling field have directly shown this using Delta null, Serrate null and/or Delta Serrate double null alleles. If these experiments cannot be performed in this specific situation, additional experiments to modulate the function of other proteins that are known to specifically regulate ligand-dependent Notch signaling such as Neuralized/Mindbomb1 can be used to strengthen the author's claim.

To further address reviewers’ concerns, instead of using Dl-DN or Kuz-DN, we have now expressed Delta-RNAi to specifically deplete Delta ligand in type II NB lineages (driven by PntP1-Gal4) or in immature INPs (driven by erm-Gal4 (II)). As shown in new Figure 3D,E, the supernumerary NB phenotype induced by *vps35* mutation was completely or largely rescued by PntP1-Gal4>Dl-IR or erm-Gal4 (II)>Dl-IR. Given that erm-Gal4 (II) is a weaker driver than insc-Gal4, as shown in Author response image 1, we concluded that Notch overactivation in vps35 mutant brains is largely, if not completely, ligand-dependent.

Also, the authors should describe the precise nature of the dominant negative reagents of Delta and Kuzbanian used in their experiments, rather than broadly labeling them as "dominant negative".

We have followed the reviewers’ suggestions and changed the text, with description of the precise nature of Delta-DN and Kuz-DN underlined here: “Type II neuroblast lineage-specific or immature INP-specific depletion of the ligand Delta as well as neuroblast lineage-specific expression of a dominant negative form of Delta (Dl-DN) that lacks its intracellular domain (Baonza et al., 2000; Flores et al., 2000; Huppert et al., 1997) completely or potently suppressed brain tumor phenotypes caused by vps35 mutations (Figure 3D,E and Figure 3—figure supplement 2A,B). Furthermore, type II neuroblast lineage-specific or immature INP-specific expression of a dominant negative form of the metalloprotease Kuzbanian (Kuz-DN), which lacks its protease activity and thereby specifically blocks ligand-induce S2 cleavage of Notch (Lieber et al., 2002; Mumm et al., 2000; Pan and Rubin, 1997), also phenocopied the effect of Notch-RNAi in inhibiting brain tumor formation (Figure 3D,E).” (Subsection “Retromer regulates retrograde trafficking of Notch receptors”).

Finally, the authors propose a model in which hypo-Ubiquitinated Notch becomes activated by ligands that are present in luminal vesicles within the multi-vesicular body. However, the authors do not provide any immunostaining data that shows that ligands are indeed present in the Rab5/7 positive enlarged endosomes. This is a simple experiment that can be done relatively quickly to support their model of "ligand-dependent" Notch activation that occurs upon loss of retromer function.

We are grateful to the reviewers for this great point. Our earlier attempts to examine Delta distribution did not work well, due to the fact that performing anti-Delta immunostaining using conventional larval brain immunostaining protocol resulted in low signal-to-noise ratio. Here we have now optimized anti-Delta immunostaining protocol (see anti-Delta immunostaining in Materials and methods section) and carried out anti-Delta immunostaining in WT versus vps35 mutant brains. As shown in new Figure 5G, Delta puncta clearly colocalized with Rab7^+^ enlarged endosomes in *vps35* mutant cells (arrowheads), strongly supporting our model in which Notch is overactivated in a ligand-dependent manner upon retromer dysfunction.

ESCRTs:The authors argue that the retromer-LOF phenotype is independent of the ESCRT pathway reported previously. Then, it is curious that Hrs-GOF (Figure 5A-C) rescued the vps35 mutant phenotype, whereas the Hrs-LOF (Figure 6G) had no effect. Given the use of PntP1-GAL4 in these experiments, the successful rescue by Hrs-GOF might be also caused by ectopic expression of Hrs in the neuroblast, rather than INP. Please address this concern with more specific manipulations.

We have now followed the reviewers’ suggestion and carried out this more specific rescue experiment. As shown in Author response image 1, immature INP-specific overexpression of Hrs, driven by erm-Gal4 (II), effectively rescued the neuroblast overproliferation phenotypes in vps35 mutants (Author response image 1). To understand why the rescuing effect of erm-Gal4 (II)>Hrs was not as complete and potent as insc-Gal4>Hrs (main Figure 5B,C), we compared the two driver lines’ ability to overexpress Hrs and thereby sort Notch receptor into lysosomes. Hrs driven by erm-Gal4 (II) exhibited much weaker effects than driven by insc-Gal4 in transporting Notch into LAMP-GFP marked lysosomes in INPs (arrowheads in Author response image 1, bottom panel). These results indicate that erm-Gal4 (II) is a much weaker driver than insc-Gal4 in INPs. Taken together, our new results led us to conclude that Hrs GOF in immature INPs is sufficient to rescue the brain tumor phenotype in vps35 mutants.

**Author response image 1. respfig1:** Retromer regulates Notch signalling in fly neuroblast lineages with high specificity. (**A,B**) Larval brain lobes of indicated genotypes were stained for Dpn and Pros. Quantification of total neuroblast number per brain lobe is shown in (**B**). **p < 0.001; (n=10-15). (**C**) Hrs driven by *erm*-Gal4 (II) showed much weaker effects in reducing type II neuroblast lineage number (upper panel) or transporting Notch receptor into LAMP-GFP^+^ lysosomes (arrowheads) in INPS (bottom panel) than driven by *insc*-Gal4. Close-up images of type II neuroblast lineages are displayed at the bottom panel. Note that type II neuroblast lineages are encircled by white dashed lines.

In addition, the authors refer to the enlarged endosomes they observe in retromer defective neuroblast-like cells as "multi-vesicular bodies" (MVBs, subsection “vps35 mutant dedifferentiating neural progenitors contained enlarged Rab7-positive endosomal vesicles” and throughout paper) where Notch accumulates and becomes activated. However, to claim that these Rab5/Rab7 positive structures are MVBs, the authors should show the presence of enlarged endosomes/vesicles that contain intraluminal vesicles through transmission electron microscopy or some sort of super-resolution light microscopy in retromer deficient cells. Otherwise, these structures can be enlarged endosomes of mixed identity that lack internal vesicles, hence not meeting the definition of MVBs. If these structures are not MVBs, or if it is not technically feasible to analyze them further within the allocated time frame, the structure should be rather called 'enlarged endosomes' and the authors model of ligand-dependent activation of Notch occurring on the limiting membrane of MVBs through ligands present in intraluminal vesicles should be toned down/amended.

We thank the reviewers for this very insightful point. We have now performed transmission electron microscopy (TEM) analysis of WT and *vps35* mutant larval brains. As shown in new Figure 2C, MVBs containing intraluminal vesicles are greatly enlarged in *vps35* mutant neuroblasts than they are in WT neuroblasts (MVBs are highlighted in purple). These TEM results, together with the presence of Delta ligand in *vps35* mutant Rab7-positive enlarged endosomes, strongly supported our model whereby ligand-dependent activation of Notch occurs at the limiting membrane of MVBs through ligands present in intraluminal vesicles.